# Neutralizing negative epigenetic regulation by HDAC5 enhances human haematopoietic stem cell homing and engraftment

Xinxin Huang[1], Bin Guo[1], Sheng Liu[2], Jun Wan[2] & Hal E. Broxmeyer[1]

Enhancement of hematopoietic stem cell (HSC) homing and engraftment is clinically critical, especially for cord blood (CB) hematopoietic cell transplantation. Here we report that specific HDAC5 inhibition highly upregulates CXCR4 surface expression in human CB HSCs and progenitor cells (HPCs). This results in enhanced SDF-1/CXCR4-mediated chemotaxis and increased homing to the bone marrow environment, with elevated SCID-repopulating cell (SRC) frequency and enhanced long-term and secondary engraftment in NSG mice. HDAC5 inhibition increases acetylated p65 levels in the nucleus, which is important for *CXCR4* transcription. Inhibition of nuclear factor-κB (NF-κB) signaling suppresses HDAC5-mediated CXCR4 upregulation, enhanced HSC homing, and engraftment. Furthermore, activation of the NF-κB signaling pathway via TNFα also results in significantly increased CXCR4 surface expression, enhanced HSC homing, and engraftment. These results demonstrate a previously unknown negative epigenetic regulation of HSC homing and engraftment by HDAC5, and allow for a new and simple translational strategy to enhance HSC transplantation.

[1] Department of Microbiology and Immunology, Indiana University School of Medicine, Indianapolis, IN 46202, USA. [2] Department of Medical and Molecular Genetics, Indiana University School of Medicine, Indianapolis, IN 46202, USA. These authors contributed equally: Xinxin Huang, Bin Guo. Correspondence and requests for materials should be addressed to H.E.B. (email: hbroxmey@iupui.edu)

Hematopoietic stem cells (HSCs) are the only cells that give rise to all blood cell lineages throughout life[1]. Allogeneic hematopoietic cell transplantation (HCT) is a life-saving therapy to treat patients with hematologic disorders and cancer[2]. Human cord blood (CB) contains a life-saving source of HSC and hematopoietic progenitor cell (HPC) for transplantation[3,4]. However, limited numbers of HSC/HPC or poor homing are problematic for efficient CB HCT[5,6]. Although extensive efforts have been devoted to ex vivo expansion of HSCs aimed at facilitating HSC engraftments and clinical applications[7–9], new insights into intrinsic and extrinsic regulation of HSC migration/homing will allow new strategies to improve HCT efficacy.

Intravenously transplanted HSCs migrate to the bone marrow (BM) niche, where they are maintained and balanced with proliferation and differentiation[10,11]. Stromal cell-derived factor-1 (SDF-1)/chemokine C-X-C receptor-4 (CXCR4) interactions are implicated as a critical axis regulating HSC trafficking and homing to the BM environment[12,13]. Modulating SDF-1/CXCR4 interactions of HSC/HPC can be used to improve the efficiency of HSC homing. For example, Prostaglandin E2 (PGE2), cyclic adenosine monophosphate, or glucocorticoid treatment facilitates HSC homing by upregulating surface CXCR4 expression[14–16], whereas DPP4/CD26 inhibition enhances HSC homing and engraftment via blockage of SDF-1 cleavage[17], and mild hyperthermia promotes CXCR4 and lipid raft aggregation to enhance HSC homing[18].

Histone deacetylases (HDACs) are erasers of acetylation from lysine residues and have important roles in many biological processes, mainly through their repressive impacts on gene transcription[19]. In mammals, HDACs comprise 18 genes that are grouped into five subfamilies (class I, IIa, IIb, III, IV) based on their sequence similarity[20]. HDAC5 belongs to class IIa HDACs, which can shuttle between the cytoplasm and nucleus, assemble into multiprotein complexes, and be responsive to various environmental stimuli[19,20]. Previous studies have reported that the functions of HDAC5 are associated with axon regeneration[21], muscle differentiation[22], angiogenesis[23], T-cell function[24], and cancer[25–28]. Of note, HDAC5-mediated deacetylation of signal transducer and activator of transcription 3 (STAT3) has been reported to regulate nuclear localization and transcriptional activity of STAT3, resulting in changes of hypothalamic leptin signaling and energy homeostasis[29]. However, the function of HDAC5 in regulating HSC has not been investigated.

In the present study, we demonstrate that specific HDAC5 inhibition leads to upregulation of CXCR4 surface expression in human CB HSCs and HPCs. Furthermore, we show that inhibition of HDAC5 results in increased SDF-1/CXCR4-mediated chemotaxis and homing, with elevated in vivo engraftment. Mechanistically, HDAC5 inhibition increases acetylated p65 levels associated with CXCR4 promoter region, whereas inhibition of nuclear factor (NF)-κB signaling suppresses both HDAC5-mediated CXCR4 upregulation and enhanced HSC homing. Moreover, activation of the NF-κB signaling pathway via tumor necrosis factor-α (TNFα) also results in significantly increased CXCR4 surface expression and enhanced HSC homing. Taken together, these results suggest that HDAC5 negatively regulates CXCR4 transcription and HSC homing via p65 acetylation. Our observations allow for a new and simple translational strategy to enhance HSC transplantation-based therapies.

## Results

### Inhibition of HDAC enhances CB HSC homing and engraftment.
We hypothesized that epigenetic regulations contribute to the expression of CXCR4 receptor and HSC homing. To identify new epigenetic regulators of CXCR4 receptor expression, we screened a chemical compound library of epigenetic enzyme inhibitors to evaluate their effects on membrane CXCR4 expression in CB CD34$^+$ cells. Treatment of CB CD34$^+$ cells for 16 h with a HDAC inhibitor, M344, strongly upregulated membrane CXCR4 expression (Fig. 1a and Supplementary Fig. 1a). Confocal imaging and flow cytometry analysis both revealed that M344 treatment strongly increased membrane CXCR4 expression compared with vehicle control (Fig. 1b–d). In addition, expression of membrane CXCR4 on CB CD34$^+$ cells was enhanced after treating cells with other HDAC inhibitors, including Vorinostat, Trichostatin A, and Belinostat (Supplementary Fig. 1b). The effect of M344 in a rigorously defined population of HSCs (CD34$^+$CD38$^-$CD45RA$^-$CD49f$^+$CD90$^+$) was associated with a 2.5-fold increase in surface CXCR4 expression (Fig. 1e). M344 also enhanced surface expression of CXCR4 on multipotent progenitors (MPPs, CD34$^+$CD38$^-$CD45RA$^-$ CD90$^-$CD49f$^-$), CD34$^+$CD38$^-$CD90$^+$, and CD34$^+$CD38$^-$ cells (Supplementary Fig. 1c-e). In contrast, the surface expression of CD49d/CD29 (VLA-4), also involved in HSC migration and homing[30], was not changed after M344 treatment (Supplementary Fig. 1f, g).

The effect of HDAC inhibition on HSC chemotaxis was evaluated in an in vitro transwell migration assay. Both vehicle and M344-treated CB CD34$^+$ cells showed significant migration to SDF-1; however, chemotaxis was notably higher in the M344-treated group (Fig. 1f). Enhanced migration to SDF-1 by M344 was also observed in the HSC population (Fig. 1g). Chemotaxis of CB CD34$^+$ cells to SDF-1 was blocked by CXCR4 antagonist AMD3100[31], suggesting that the enhanced migration was mediated through CXCR4 (Fig. 1f). To evaluate in vivo homing, vehicle and M344-treated CB CD34$^+$ cells were injected into sublethally irradiated NSG mice and human cell homing to mouse BM and spleen were analyzed 24 h after transplantation. M344 treatment enhanced CB CD34$^+$ cells homing in both BM and spleen in NSG mice (Fig. 1h).

Long-term engraftment of human CB CD34$^+$ cells was assessed by limiting dilution assay to compare frequencies of SCID-repopulating cells (SRCs) in vehicle and M344-treated CB CD34$^+$ cells. M344-treated CB CD34$^+$ cells manifested significantly greater engraftment in primary NSG recipient mice compared with that of the vehicle control-treated group (Fig. 2a, b). Human myeloid and B-cell chimerisms were also significantly increased (Fig. 2b). Poisson distribution analysis revealed an SRC frequency of 1/3216 for the vehicle control-treated group and 1/746 for the M344-treated group. This reflected respectively 311 and 1341 SRCs in $1 \times 10^6$ cells from vehicle control and M344-treated cultures, resulting in a 4.3-fold increase in the number of functionally detectable SRCs compared with the vehicle control group (Fig. 2c, d and Supplementary Table 1, 2). To evaluate self-renewal capability, we transplanted BM cells from primary NSG recipient mice into secondary sublethally irradiated NSG mice. The M344-treated group resulted in greater engraftment in transplanted secondary recipients (Fig. 2e), suggesting we had increased engraftment of long-term self-renewing human CB HSCs after M344 treatment.

### HDAC5 inhibition promotes CB HSC homing and engraftment.
Eighteen HDACs divided into four classes have been identified in humans[20,32]. Mechanisms regarding HDAC regulation of HSC homing and engraftment are largely unknown. Using short hairpin RNA to knockdown expression of individual HDACs in CB CD34$^+$ cells (Supplementary Fig. 2), we surprisingly found that only HDAC5 shRNA transfection resulted in upregulation of membrane CXCR4 expression (Fig. 3a). Flow cytometry and confocal imaging analysis revealed increased membrane CXCR4 expression by LMK235[33], a selective inhibitor

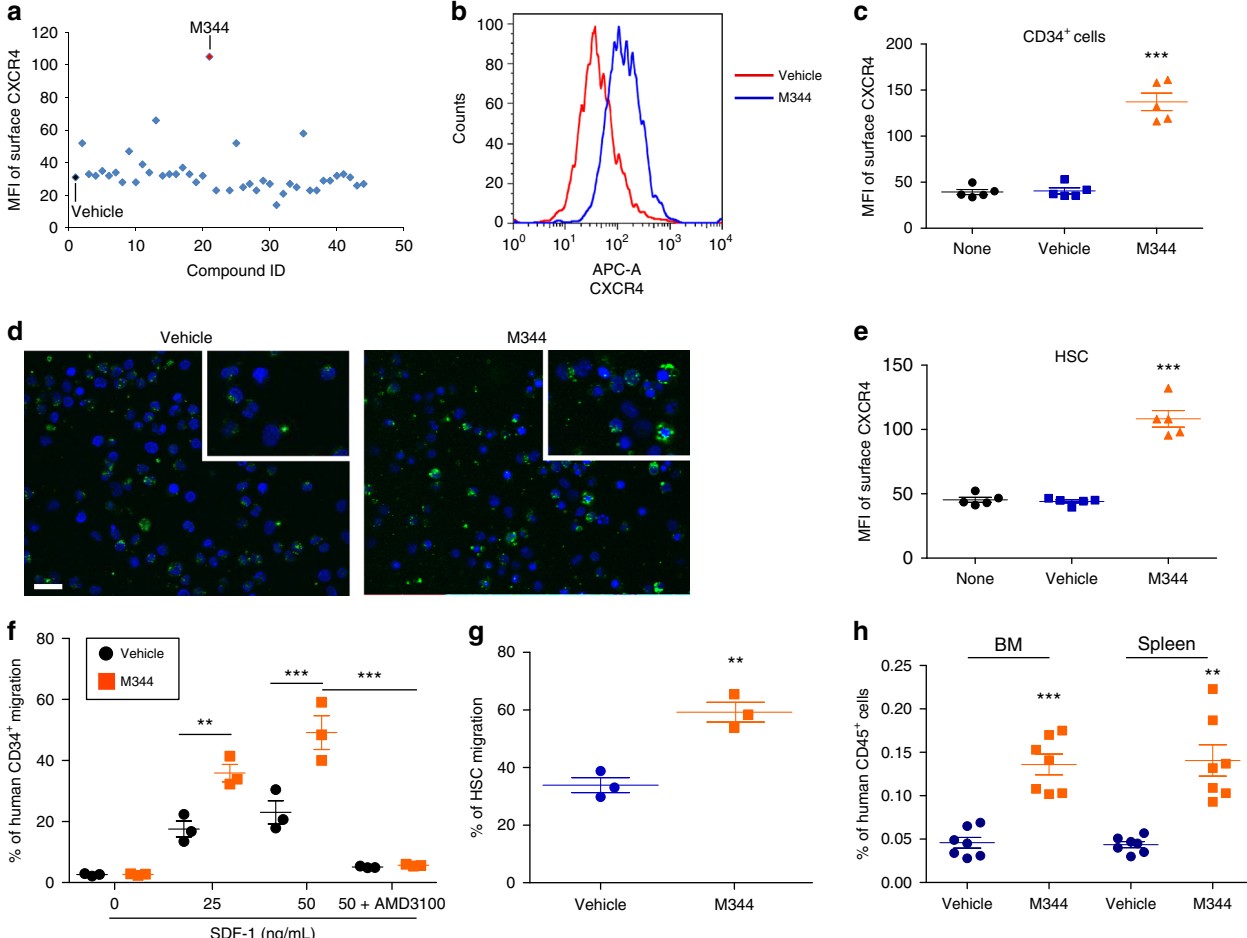

**Fig. 1** Inhibition of HDAC upregulates surface CXCR4 expression and promotes homing of human hematopoietic stem and progenitor cells. **a** Mean fluorescence intensity (MFI) of surface CXCR4 of human CB CD34$^+$ cells after treating cells for 16 h with compounds (1 μM) from the SCREEN-WELL Epigenetics Library. One compound, BML-266, was excluded from the results because of unspecific autofluorescence. **b** Histogram of surface CXCR4 expression of human CB CD34$^+$ cells treated with vehicle or HDAC inhibitor M344. Representative histogram from five independent experiments is shown. **c** Quantification of mean fluorescence intensity (MFI) of surface CXCR4 of human CB CD34$^+$ cells treated with vehicle or HDAC inhibitor M344. None indicates the group without any treatment. Data pooled from five independent experiments are shown ($n = 5$, one-way ANOVA, ***$p < 0.001$). **d** Confocal imaging analysis of surface CXCR4 expression of human CB CD34$^+$ cells treated with vehicle or M344. FITC (green) indicates CXCR4 expression; DAPI (blue) labels the cell nucleus. Representative images from two independent experiments are shown (the inset shows the amplified part of the image). Scale bar: 20 μm. **e** Quantification of mean fluorescence intensity (MFI) of surface CXCR4 of human CB HSCs (CD34$^+$CD38$^-$CD45RA$^-$CD90$^+$CD49f$^+$) treated with vehicle or M344. None indicates the group without any treatment. Data pooled from five independent experiments are shown ($n = 5$, one-way ANOVA, ***$p < 0.001$). **f** Chemotaxis of human CB CD34$^+$ cells toward human recombinant SDF-1, as quantified by flow cytometry. Data pooled from three independent experiments are shown (each dot represents an independent chemotaxis, $n = 3$ CB samples, one-way ANOVA, **$p < 0.01$, ***$p < 0.001$). **g** Migration of human phenotypic HSCs in CB CD34$^+$ cells toward human recombinant SDF-1 (50 ng/mL), as quantified by flow cytometry. The migration percentage of HSCs was calculated by analyzing the HSC (CD34$^+$CD38$^-$CD45RA$^-$CD90$^+$CD49f$^+$) frequency using flow cytometry. Data pooled from three independent experiments are shown (each dot represents an independent chemotaxis, $n = 3$ CB samples, $t$-test, **$p < 0.01$). **h** The percentage of human CD45$^+$ cells in the BM and spleen of NSG mice 24 h after transplantation with 500,000 CB CD34$^+$ cells that had been treated with vehicle or M344. Data pooled from seven independent experiments are shown ($n = 7$ mice per group, $t$-test, **$p < 0.01$, ***$p < 0.001$). For all panels, data are shown as dot plots (mean ± SEM)

of HDAC5 (Fig. 3b–d and Supplementary Fig. 3a). In contrast, inhibition of other HDACs showed no effects on membrane CXCR4 expression in CB CD34$^+$ cells (Supplementary Fig. 3b). Similar to M344, HDAC5 inhibition by LMK235 resulted in significantly higher membrane CXCR4 expression in CB HSCs, MPPs, CD34$^+$CD38$^-$CD90$^+$, and CD34$^+$CD38$^-$ cells (Supplementary Fig. 3c-f). We also examined the expression of HDAC5 and found that HDAC5 is highly expressed in CB HSCs (Supplementary Fig. 3g, h), suggesting a potential important role of HDAC5 in HSCs.

We next assessed whether HDAC5 inhibition can promote chemotaxis of CB HSCs toward SDF-1. LMK235 treatment significantly enhanced chemotaxis of CB CD34$^+$ cells toward SDF-1, which can be fully blocked by CXCR4 antagonist AMD3100 (Fig. 3e). Enhanced migration to SDF-1 by LMK235 was also observed in the HSC population (Fig. 3f). We then tested whether HDAC5 inhibition enhanced homing in vivo. Human CB CD34$^+$ cells were treated with vehicle and LMK235 for 16 h and transplanted into sublethally irradiated NSG mice, we found that LMK235 treatment greatly increased the homing efficiency of CB CD34$^+$ cells (Fig. 3g). To further demonstrate that the homed cells contain self-renewing HSCs, we collected BM cells from primary NSG recipients used in the homing assay and injected them into secondary recipients to check for long-term

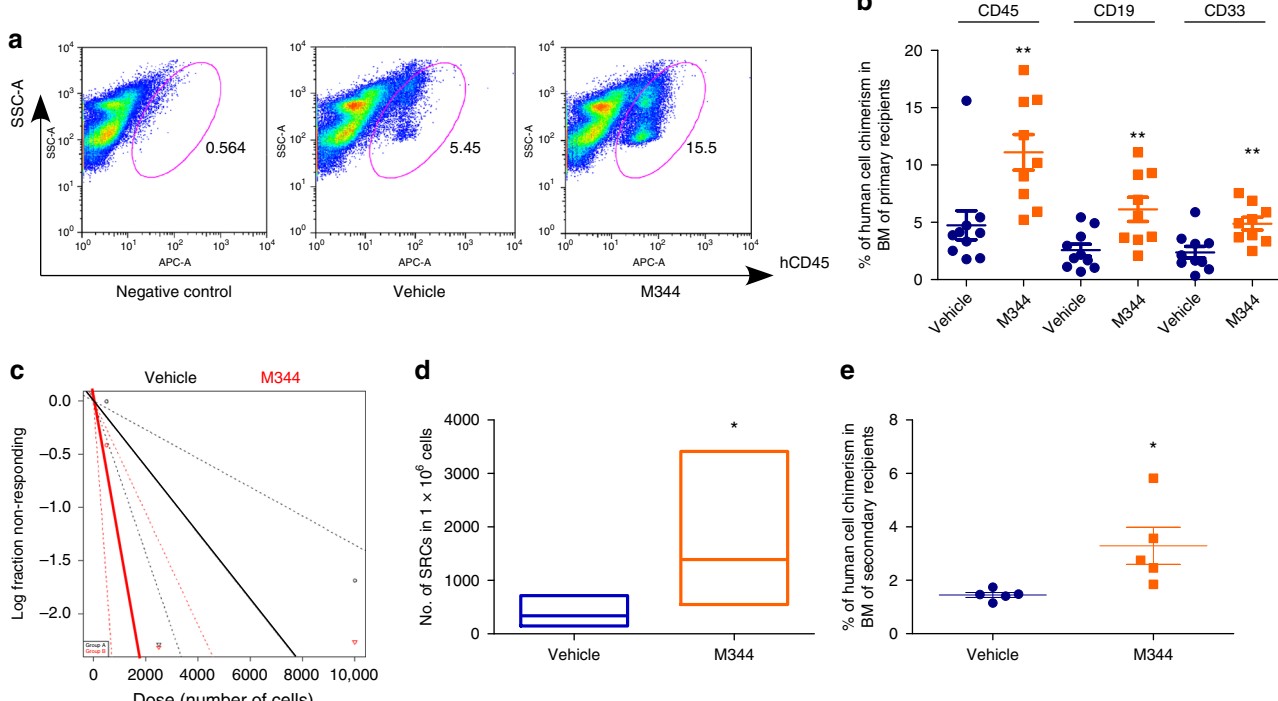

**Fig. 2** Inhibition of HDAC promotes long-term engraftment of human CB HSC. **a** Representative flow cytometric analysis of human engraftment in the BM of NSG mice, 16 weeks after transplantation. Left is from a mouse without transplantation (negative control), center and right are from mice transplanted with human CB CD34+ cells treated with vehicle control or M344 for 16 h. Human engraftment was assessed as the percentage of human CD45+ cells. **b** The percentage of human CD45+ cells, B-cell (CD19+), and myeloid cell (CD33+) chimerism in the BM of NSG mice after transplantation with 10,000 CB CD34+ cells that had been treated with vehicle or M344. The data were pooled from two independent experiments ($n = 10$ mice for vehicle group, $n = 9$ for M344 group, t-test, **$p < 0.01$). **c**, **d** The frequency of human SRCs in CB CD34+ cells treated with vehicle control or M344. **c** Poisson statistical analysis of data from Supplementary Table 1. $n = 30$ mice in total. Shapes represent the percentage of negative mice for each dose of cells. Solid lines indicate the best-fit linear model for each data set. Dotted lines represent 95% confidence intervals. **d** HSC frequencies (line in the box) and 95% confidence intervals (box) presented as the number of SRCs in $1 \times 10^6$ CD34+ cells. Poisson statistical analysis, *$p < 0.05$. **e** Human CD45+ cell chimerism in the BM of secondary recipient NSG mice at 16 weeks, which had been transplanted with $5 \times 10^6$ BM cells from primary recipient NSG mice ($n = 5$ mice per group, t-test, *$p < 0.05$). For all panels, data are shown as dot plots (mean ± SEM)

engraftment. We found that LMK235-treated CD34+ cells also showed increased engraftment in these transplanted secondary recipients (Fig. 3h), suggesting LMK235 treatment promoted homing of self-renewing CB HSCs.

Next we performed limiting dilution assay to compare the frequency of SRCs for vehicle and LMK235-treated CB CD34+ cells. Engraftment of LMK235-treated CB CD34+ cells was largely elevated in primary NSG recipient mice, compared with the vehicle control-treated group (Fig. 4a, b). Human myeloid and B-cell chimerisms were also significantly increased (Fig. 4b). The SRC frequency was 1/7916 in vehicle control-treated CB CD34+ cells with an increased SRC frequency of 1/1326 in LMK235-treated cells, with 126 and 754 SRCs, respectively, in $1 \times 10^6$ cells from vehicle control- and LMK235-treated cells, demonstrating a 6.0-fold increase in numbers of functionally detectable SRCs compared with vehicle control (Fig. 4c, d and Supplementary Tables 3, 4). LMK235-treated CB CD34+ cells also showed increased engraftment in transplanted secondary recipients (Fig. 4e). Thus, HDAC5 inhibition enhances engraftment of long-term repopulating, self-renewing human CB HSCs.

**HDAC5 inhibition promotes *CXCR4* transcription.** HDACs regulate gene transcription by modifying histone and chromatin structures[20]. Quantitative reverse transcription-PCR showed increased *CXCR4* messenger RNA levels in both M344 and LMK235-treated CB CD34+ cells compared with vehicle control

(Fig. 5a). The levels of *CXCR4* mRNA also increased in HDAC5 knockdown CB CD34+ cells, indicating that HDAC5 inhibition promotes *CXCR4* expression at a transcriptional level (Fig. 5b). Chromatin immunoprecipitation (ChIP) assays examined chromatin status at the *CXCR4* promoter region. Acetylation levels on histone 3 lysine 9 (H3K9) and histone 4 lysine 16 (H4K16) were significant higher in both M344- and LMK235-treated CB CD34+ cells compared with vehicle control, suggesting that increased histone acetylation levels at the *CXCR4* promoter region contributes to increased *CXCR4* transcription (Fig. 5c, d). The action of histone acetyltransferase is opposite to that of HDACs[34], so we tested effects of histone acetyltransferase p300 inhibitor, C646, on surface expression of CXCR4 in CB CD34+ cells. Treatment of CB CD34+ cells with C646 suppressed effects of M344 and LMK235 on CXCR4 expression (Fig. 5e). Another p300 inhibitor, EML425 showed similar suppressive effects on CXCR4 surface expression in CB CD34+ cells treated with M344 and LMK235 (Supplementary Fig. 4a). This suggests that a balance between histone acetyltransferase and deacetylase is important for CXCR4 expression in CB CD34+ cells.

**Transcriptome analysis after HDAC5 inhibition.** To gain further insight into the effects of HDAC5 inhibition, we performed RNA sequencing (RNA-seq) on LMK235- and vehicle-treated CB CD34+ cells. As expected, CXCR4 was identified as one of the positive hits (Fig. 5f). Consistent with the reports showing that

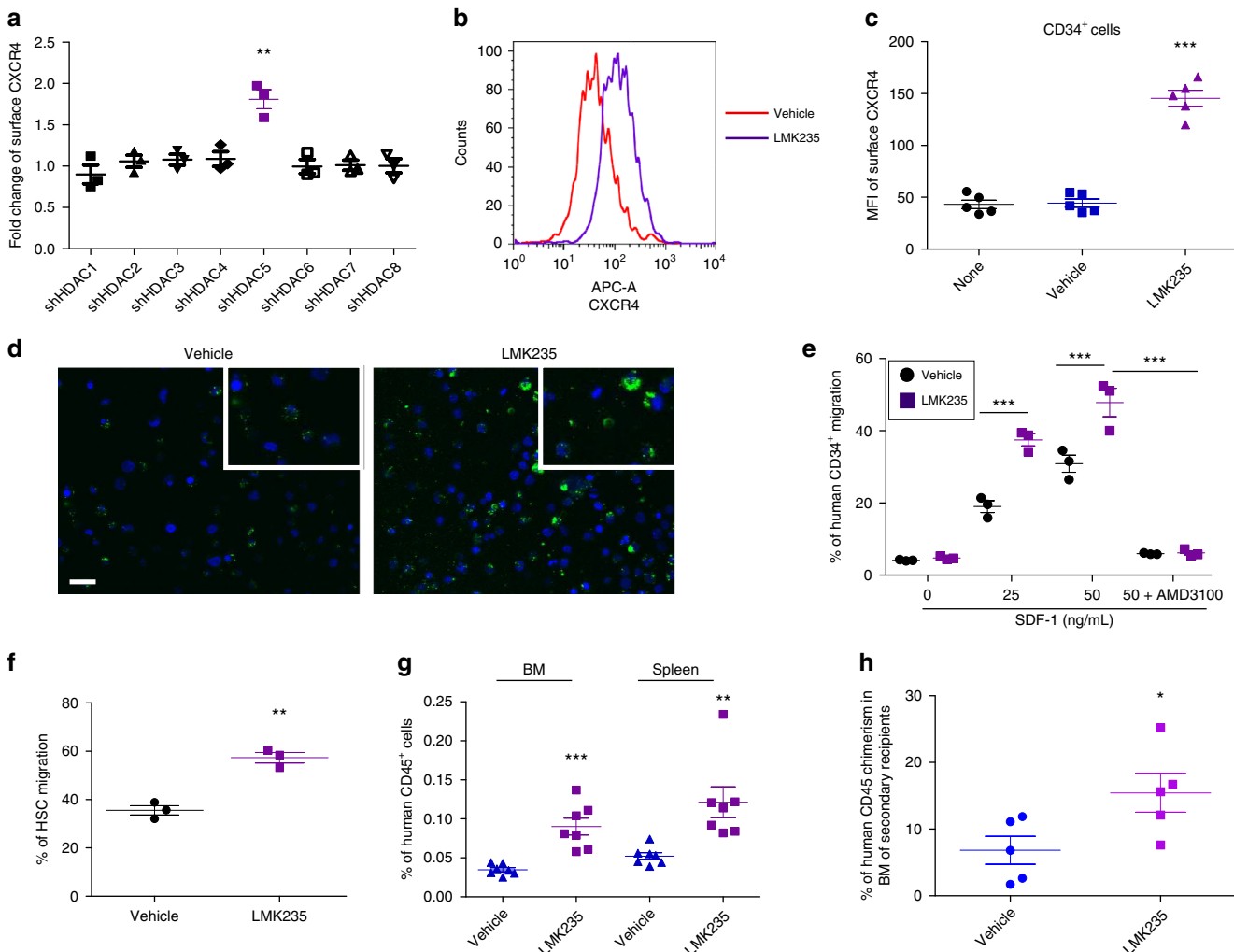

**Fig. 3** Inhibition of HDAC5 enhances CB HSC homing by upregulating CXCR4 surface expression. **a** Fold change of mean fluorescence intensity of surface CXCR4 of human CB CD34[+] cells after transfecting cells with indicated shRNA for individual HDACs. Individual shRNA knockdown efficiency was shown in Supplementary Fig. 2. Data pooled from three independent experiments are shown ($n = 3$, t-test, **$p < 0.01$). **b** Histogram of surface CXCR4 expression of human CB CD34[+] cells treated with vehicle or HDAC5 inhibitor LMK235. Representative histogram from five independent experiments is shown. **c** Quantification of mean fluorescence intensity (MFI) of surface CXCR4 of human CB CD34[+] cells treated with vehicle or HDAC5 inhibitor LMK235. None indicates the group without any treatment. Data pooled from five independent experiments are shown ($n = 5$, one-way ANOVA, ***$p < 0.001$). **d** Confocal imaging analysis of surface CXCR4 expression of human CB CD34[+] cells treated with vehicle or LMK235. Green indicates CXCR4 expression; DAPI (blue) labels the cell nucleus. Representative images from two independent experiments are shown (the inset shows the amplified part of the image). Scale bar: 20 μm. **e** Human CB CD34[+] cells migration toward human recombinant SDF-1, as quantified by flow cytometry. Data pooled from three independent experiments are shown (each dot represents an independent chemotaxis, $n = 3$ CB samples, one-way ANOVA, ***$p < 0.001$). **f** Migration of human phenotypic HSCs in CB CD34[+] cells toward human recombinant SDF-1 (50 ng/mL), as quantified by flow cytometry. Data pooled from three independent experiments are shown (each dot represents an independent chemotaxis, $n = 3$ CB samples, t-test, **$p < 0.01$). **g** The percentage of human CD45[+] cells in the BM and spleen of NSG mice 24 h after transplantation with 500,000 CB CD34[+] cells that had been treated with vehicle or LMK235. Data pooled from seven independent experiments are shown ($n = 7$ mice per group, t-test, **$p < 0.01$, ***$p < 0.001$). **h** The percentage of human CD45[+] cells in the BM of secondary recipient NSG mice that were transplanted with BM cells from primary mice used in the homing assay ($n = 5$ mice per group, t-test, *$p < 0.05$). For all panels, data are shown as dot plots (mean ± SEM)

PGE2 promotes HSC homing[15], we identified that PTGER4, a PGE2 receptor, has a strong enrichment in LMK235-treated CB CD34[+] cells (Fig. 5f). Gene ontology (GO) analysis revealed an upregulation of gene sets linked with cell locomotion, cell motility, cell migration, and cell adhesion, all of which have been reported to be involved in HSC homing (Fig. 5g). Cytoskeleton rearrangements have been shown to be crucial during HSC homing process[35,36]. GO analysis also showed a significant enrichment of transcripts associated with cytoskeleton and cytoskeleton-binding proteins in LMK235-treated CB CD34[+] cells (Fig. 5g). These transcriptomic data provide additional evidence supporting that HDAC5 modulates HSC homing through transcriptional regulation.

**HDAC5 regulates levels of p65 acetylation.** NF-κB signaling and NF-κB subunit p65 regulate CXCR4 expression in breast cancer cells and neuroblastoma[37,38], and acetylation of p65 at lysine 310 (K310) regulates its DNA-binding activity and target gene transcription[39,40]. We hypothesized that HDAC5 regulates p65 acetylation to promote CXCR4 expression and HSC homing. We then examined the levels of p65 acetylation and found that p65

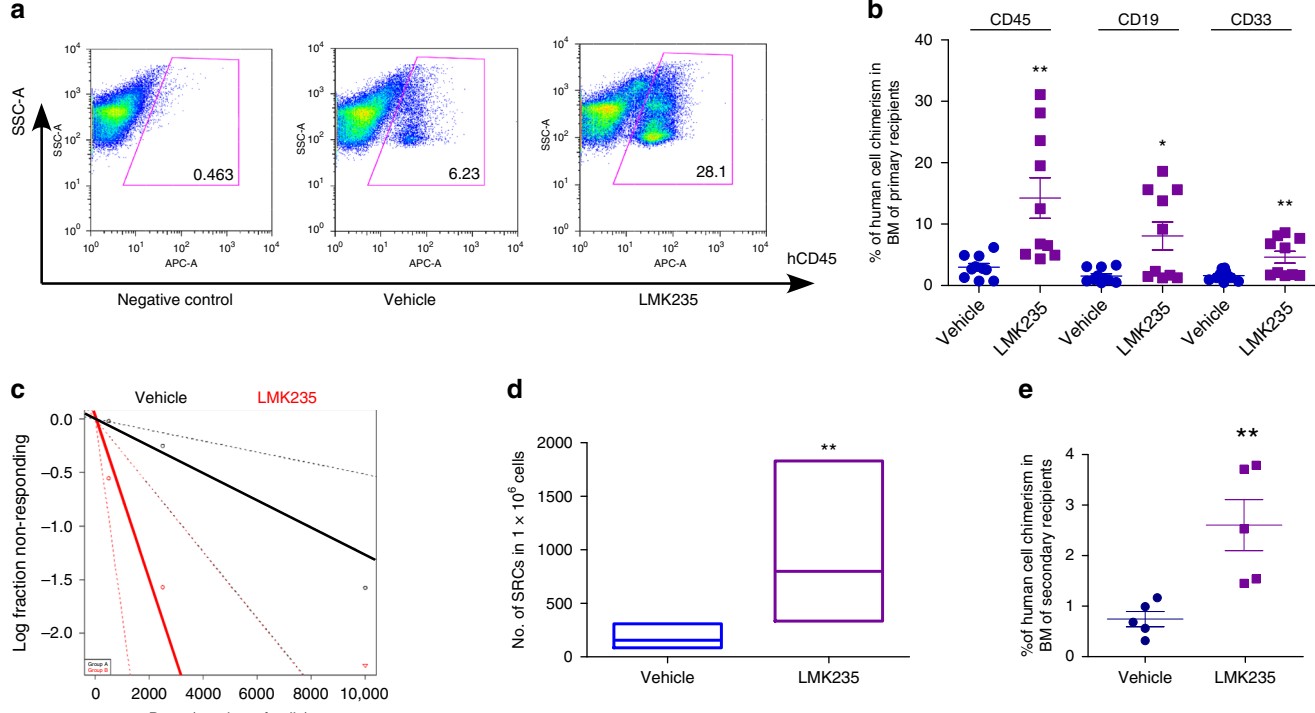

**Fig. 4** Inhibition of HDAC5 increases long-term engraftment of human CB HSC. **a** Representative flow cytometric analysis of human engraftment in the BM of NSG mice, 16 weeks after transplantation. Left is from a mouse without transplantation (negative control), center and right are from mice transplanted with human CB CD34$^+$ cells treated with vehicle control or LMK235 for 16 h. Human engraftment was assessed as the percentage of human CD45$^+$ cells. **b** The percentage of human CD45$^+$ cells, B-cell (CD19$^+$), and myeloid cell (CD33$^+$) chimerism in the BM of NSG mice after transplantation with 10,000 CB CD34$^+$ cells that had been treated with vehicle or LMK235. The data were pooled from two independent experiments ($n = 10$ mice per group, $t$-test, *$p < 0.05$. **$p < 0.01$). **c,d** The frequency of human SRCs in CB CD34$^+$ cells treated with vehicle control or LMK235. **c** Poisson statistical analysis of data from Supplementary Table 3. $n = 30$ mice in total. Shapes represent the percentage of negative mice for each dose of cells. Solid lines indicate the best-fit linear model for each data set. Dotted lines represent 95% confidence intervals. **d** HSC frequencies (line in the box) and 95% confidence intervals (box) presented as the number of SRCs in $1 \times 10^6$ CD34$^+$ cells. **$p < 0.01$. **e** Human CD45$^+$ cell chimerism in the BM of secondary recipient NSG mice at 16 weeks, which had been transplanted with $5 \times 10^6$ BM cells from primary recipient NSG mice ($n = 5$ mice per group, $t$-test, **$p < 0.01$). For all panels, data are shown as dot plots (mean ± SEM)

acetylation at K310 was markedly increased in CB CD34$^+$ cells with M344 and LMK235 treatment (Fig. 6a, b). Treatment of CB CD34$^+$ cells with C646 suppressed effects of M344 and LMK235 on p65 acetylation (Fig. 6b). Confocal imaging analysis confirmed elevated p65 acetylation in the nucleus of CB CD34$^+$ cells with M344 and LMK235 treatment (Fig. 6c). p65 expression levels were unaltered in M344 and LMK235 compared with vehicle control-treated CB CD34$^+$ cells (Supplementary Fig. 4b), suggesting that regulation of p65 by HDAC5 is posttranslational. Using ChIP assay we detected increased levels of acetylated p65 binding to the *CXCR4* promoter region in the LMK235-treated group compared with vehicle control (Fig. 6d). We next investigated whether HDAC5 physically interacted with p65. Indeed, we found that HDAC5 immunoprecipitated with acetylated p65 (Fig. 6e). Taken together, these results suggest HDAC5 interacts with p65 and regulates its acetylation, which potentially upregulates *CXCR4* transcription.

**Inhibition of NF-κB signaling suppresses HSC homing**. To further confirm the role of NF-κB signaling in HSC homing, we next evaluated effects of NF-κB inhibition on surface expression of CXCR4 and CB HSC homing. Inhibition of the NF-κB signaling pathway by Andrographolide, BMS345541, or Pyrrolidinedithiocarbamate ammonium (PDTC) repressed effects of M344 and LMK235 on CXCR4 upregulation (Fig. 7a, b and Supplementary Fig. 5a). Andrographolide or BMS345541 treatment also suppressed SDF-1/CXCR4-mediated CB CD34$^+$ cell chemotaxis

(Fig. 7c and Supplementary Fig. 5b). Next we assessed the effects of NF-κB inhibition in vivo. Inhibition of the NF-κB signaling pathway by Andrographolide blocked enhanced homing of LMK235-treated CB CD34$^+$ cells (Fig. 7d and Supplementary Fig. 5c). We followed long-term engraftment and confirmed that Andrographolide also suppressed LMK235-mediated increases of human cell chimerism in primary recipient mice (Fig. 7e, f). Our data indicate that inhibition of NF-κB signaling suppresses the effects of HDAC5 inhibition on CXCR4 upregulation, increased HSC homing, and engraftment.

**TNFα treatment results in enhanced HSC homing**. TNFα activates canonical NF-κB signaling[41]. We found significant upregulation of surface CXCR4 expression after TNFα treatment in CB CD34$^+$ cells, effects suppressed by inhibitors of the NF-κB signaling pathway (Fig. 8a). The levels of *CXCR4* mRNA also increased after TNFα treatment in CB CD34$^+$ cells (Supplementary Fig. 6a). Moreover, TNFα treatment significantly enhanced chemotaxis towards SDF-1 and in vivo homing (Fig. 8b, c). Inhibition of the NF-κB pathway by Andrographolide blocked effects of TNFα-enhanced homing in vivo (Fig. 8c and Supplementary Fig. 6b). Long-term engraftment of human CB CD34$^+$ cells was assessed in primary recipients transplanted with vehicle and TNFα-treated CB CD34$^+$ cells. TNFα-treated CB CD34$^+$ cells showed significantly elevated engraftment in NSG mice compared with that of the vehicle control-treated group (Fig. 8d, e). Human myeloid and B-cell

chimerisms were also increased (Fig. 8e). Thus, activation of NF-κB signaling is sufficient to promote surface expression of CXCR4 and enhance homing and engraftment of CB CD34$^+$ cells.

## Discussion

HSC homing efficiency is a pivotal factor for a successful HCT-based clinical therapy. In this work, we show that upon HDAC5 inhibition, acetylated p65 binds to the CXCR4 promotor region,

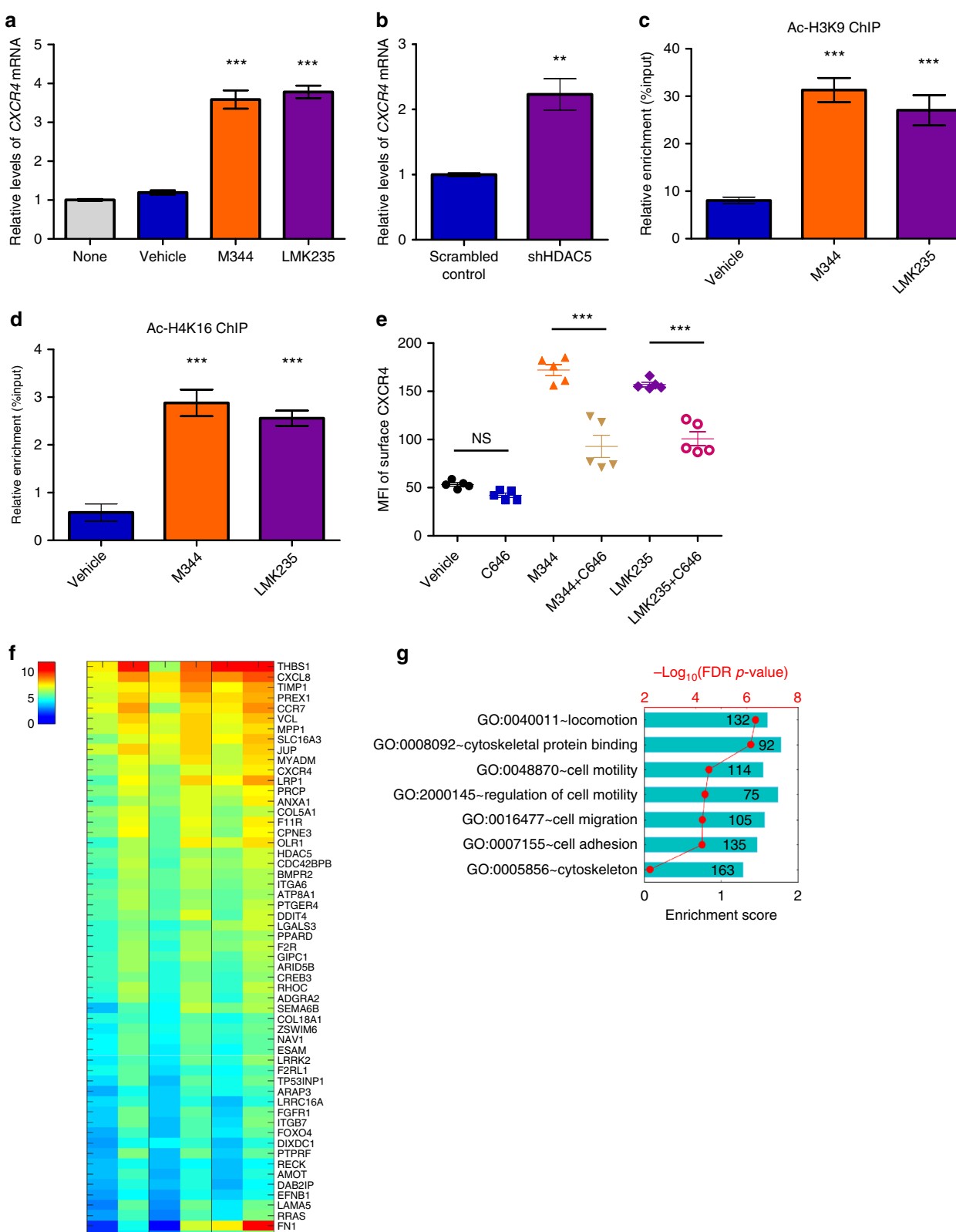

transcriptionally upregulates CXCR4 surface expression and enhances homing and long-term engraftment of self-renewing human CB HSCs (Fig. 9). Our study could lead to significant improvements particularly for CB transplantation, where limited HSC numbers presented in each single unit create clinical limitations.

It has been reported that HDAC inhibition promotes HSC ex vivo expansion and enhances engraftment[42,43]. However, the role of HDAC in HSC homing and the underlying mechanism were largely unknown. We now demonstrate, for the first time, that HDAC5 inhibition facilitates HSC engraftment by upregulating surface CXCR4, enhancing chemotaxis to SDF-1 and homing to the BM. HDAC inhibition has been shown to alter CXCR4 expression in various cell types and it seems the regulation could be either positive or negative depending on cell context. HDAC inhibition decreased surface CXCR4 expression in dendritic cells, CD4 T cells, and monocytes[44,45], whereas inhibition of HDAC promoted surface CXCR4 expression in mesenchymal stem cells[46] and HSCs (this study). It would be interesting to explore what kind of cell-type-specific factors cooperated with which specific HDAC to regulate CXCR4 expression in various types of cells.

HDAC5 belongs to the class IIa HDAC family and has been known to shuttle between the cytosol and the nucleus[20]. HDAC5 phosphorylation can be regulated by external signals, which would promote HDAC5 shuttling to the cytosol and relieve its transcription suppressive function. Thus, the regulation of HDAC5 provides a mechanism for linking extracellular signaling with HSC migration/homing to the BM environment. In the original work by Marek et al.[33], LMK235 showed the highest activity at HDAC5 in the nanomolar range. However, they used in vitro activity assay to detect the efficiency of LMK235. To target the intracellular HDAC5 when treating cells, we believe that higher concentration of LMK235 will be needed. Also different cell types will be different in their capability to take up the compound. We performed titration on LMK235 for human CB CD34+ cells and determined that 1 μM worked best (Supplementary Fig 3a). Thus, we used 1 μM to treat human CB CD34+ cells for in vivo and mechanistic studies.

NF-κB signaling pathway is well documented in inflammatory responses and TNFα is an inflammatory cytokine that activates NF-κB signaling[41]. Our data show for the first time that NF-κB signaling is involved in HSC homing to the BM microenvironment. Inhibition of NF-κB signaling suppressed the effects of HDAC5 inhibitor on CXCR4 upregulation and enhanced HSC homing, whereas activation of NF-κB signaling by TNFα promoted CXCR4 expression and HSC homing. The crosstalk between NF-κB signaling and HSC homing has a physiologic implication that HSC could be protected by activating NF-κB signaling to migrate to the BM niche during inflammation and infection. Furthermore, it has been reported that genetic deletion of NF-κB subunit p65 severely impairs HSC function[47]. Consistently, we found that p65 is involved in CXCR4 transcription in CB HSC and mediated HDAC5 function.

Collectively, our findings have significant translational implications. Short-term ex vivo treatment of cells with M344, LMK235, or TNFα has the potential to greatly improve the clinical efficacy of CB transplantation, especially for the use of single CB unit for adult transplantation. Moreover, as higher expression of CXCR4 by cancer cells is associated with metastatic migration[48], our study suggests that it might be clinically feasible to inhibit tumor cell migration by targeting HDAC5.

## Methods

**Mice**. Six to eight weeks old NSG (NOD.Cg-Prkdc[scid] IL2rg[tm1Wjl]/Sz) mice were obtained from the In Vivo Therapeutics Core of the Indiana University School of Medicine (IUSM). All experimental protocols with mice were approved by The Institutional Animal Care and Use Committee of IUSM.

**Isolation and culture of CB CD34+ cells**. Isolation of CD34+ cells from normal human CB samples (CordUse, Orlando, FL, USA) was performed by density gradient centrifugation over Ficoll Paque Plus (GE Healthcare, Piscataway, NJ, USA) and collection through an human CD34 MicroBead Kit (Miltenyi Biotec, San Diego, CA, USA)[9,16,49]. CD34+ cells were then cultured in RPMI-1640 medium supplemented with 10% fetal bovine serum, 100 ng/mL stem cell factor (7466-SC-010/CF, R&D Systems, Minneapolis, MN, USA), thrombopoietin (288−TP-200/CF, R&D Systems), and Fms-like tyrosine kinase 3 ligand (Flt3L) (# 710802, BioLegend). For small molecule compound screen, 50,000 CD34+ cells were cultured in the above medium with compounds (1 μM) from the SCREEN-WELL Epigenetics Library (Enzo Life Sciences, Farmingdale, NY, USA) for 16 h. For treatment of HDAC inhibitors, the cells were incubated with M344 (S2779, 1 μM) or LMK235 (S7569, 1 μM) (Selleck Chemicals, Houston, TX, USA) for 16 h unless otherwise stated. The following compounds were also used: C646 (Selleck Chemicals, 50 μM), EML425 (Tocris Bioscience, 50 μM), Andrographolide (Tocris Bioscience, 10 μM), BMS345541 (Tocris Bioscience, 10 μM), and PDTC (Tocris Bioscience, 20 μM). Dimethyl sulfoxide (D2650, Sigma-Aldrich, St. Louis, MO, USA) was used as vehicle control.

**Immunostaining and flow cytometry**. Samples were washed and stained with antibodies at 4 °C for 30 min. Then cells were washed with 1 mL phosphate-buffered saline (PBS) and resuspended in 1% formaldehyde buffer before analysis. For intracellular staining, cells were first fixed and permeabilized using a Cell Permeabilization Kit (BD Biosciences, San Jose, CA, USA) before incubating with primary and secondary antibodies. The following antibodies from BD Bioscience were used for cell surface staining in 1:100 dilution: CD34-FITC (581), CD38-PE (HIT2), CXCR4-APC (12G5), CD45RA−PE-CF594 (HI100), CD49f-PerCPcy5.5 (GoH3), CD90-PEcy7 (5E10), CD49d-APC (9F10), CD29-PE (MAR4), CD45-APC (HI30), CD19-PE (HIB19), and CD33-PEcy7 (WM53). The following antibodies were used for intracellular staining in 1:100 dilution: anti-p65 acetyl K310 (ab19870, Abcam, Cambridge, MA, USA), anti-p65 (ab32536, Abcam). Flow cytometry analysis was performed on an LSRII flow cytometer (BD Biosciences).

**shRNA knockdown assay**. For HDAC knockdown, shRNA expression plasmids (psi-LVRU6GP, obtained from Genecopeia) targeting HDAC1-HDAC8 were transfected into CB CD34+ cells by nucleofection (Amaxa™ Nucleofector™ Technology) following Lonza's optimized protocols (Lonza, VPA-1003). After transfection, cells were cultured for 48 h and CXCR4 expressions were determined by surface staining and followed by flow cytometry analysis.

---

**Fig. 5** Inhibition of HDAC5 promotes *CXCR4* transcription. **a** *CXCR4* mRNA levels in M344- or LMK235-treated human CB CD34+ cells, relative to vehicle-treated cells, as assessed by quantitative RT-PCR. None indicates the group without any treatment. Data pooled from two independent experiments are shown ($n = 6$, one-way ANOVA, ***$p < 0.001$). **b** Human CB CD34+ cells were transfected with scrambled control or HDAC5 shRNA plasmids, then *CXCR4* mRNA levels were analyzed by quantitative RT-PCR. Data pooled from two independent experiments are shown ($n = 6$, t-test, **$p < 0.01$). **c, d** Acetylated H3K9 (Ac-H3K9, **b**) and H4K16 (Ac-H4K16, **c**) levels on *CXCR4* promoter in vehicle, M344-, or LMK235-treated human CB CD34+ cells, as assessed by ChIP assays. Data pooled from two independent experiments are shown ($n = 6$, one-way ANOVA, ***$p < 0.001$). **e** Mean fluorescence intensity (MFI) of surface CXCR4 in vehicle, C646 (50 μM), M344-, M344 + C646-, LMK235-, LMK235 + C646-treated human CB CD34+ cells, as assessed by flow cytometry. Data pooled from two independent experiments are shown ($n = 5$, one-way ANOVA, ***$p < 0.001$, NS indicates $p > 0.05$). **f** Heatmap of upregulated differentially expressed genes (DEGs) associated with "cell migration" whose average count-per-million (CPM) larger than 8 in vehicle control group (details described in Methods section). D1, D2, D3 represent three vehicle control-treated groups; K1, K2, K3 represent three LMK235-treated groups ($n = 3$ CB samples). **g** Selected GOs that are significantly enriched and showed upregulated expression in LMK235-treated group compared with vehicle control group. For all panels, data are shown as mean ± SEM

**Confocal imaging**. M344-, LMK235-, or vehicle-treated CB CD34$^+$ cells were placed on a Nunc glass dish (150680, Thermo Fisher Scientific) coated with poly-L-lysine. Immunostainings were performed using anti-CXCR4 (ab124824) or anti-p65 acetyl K310 (ab19870, Abcam) in 1:100 dilution at room temperature for 20 min followed by fluorecein isothiocyanate (FITC)-labeled secondary antibodies. Samples were washed with PBS and resuspended with buffer containing DAPI (Vector Laboratories), and visualized using an Olympus FV-1000 confocal microscope.

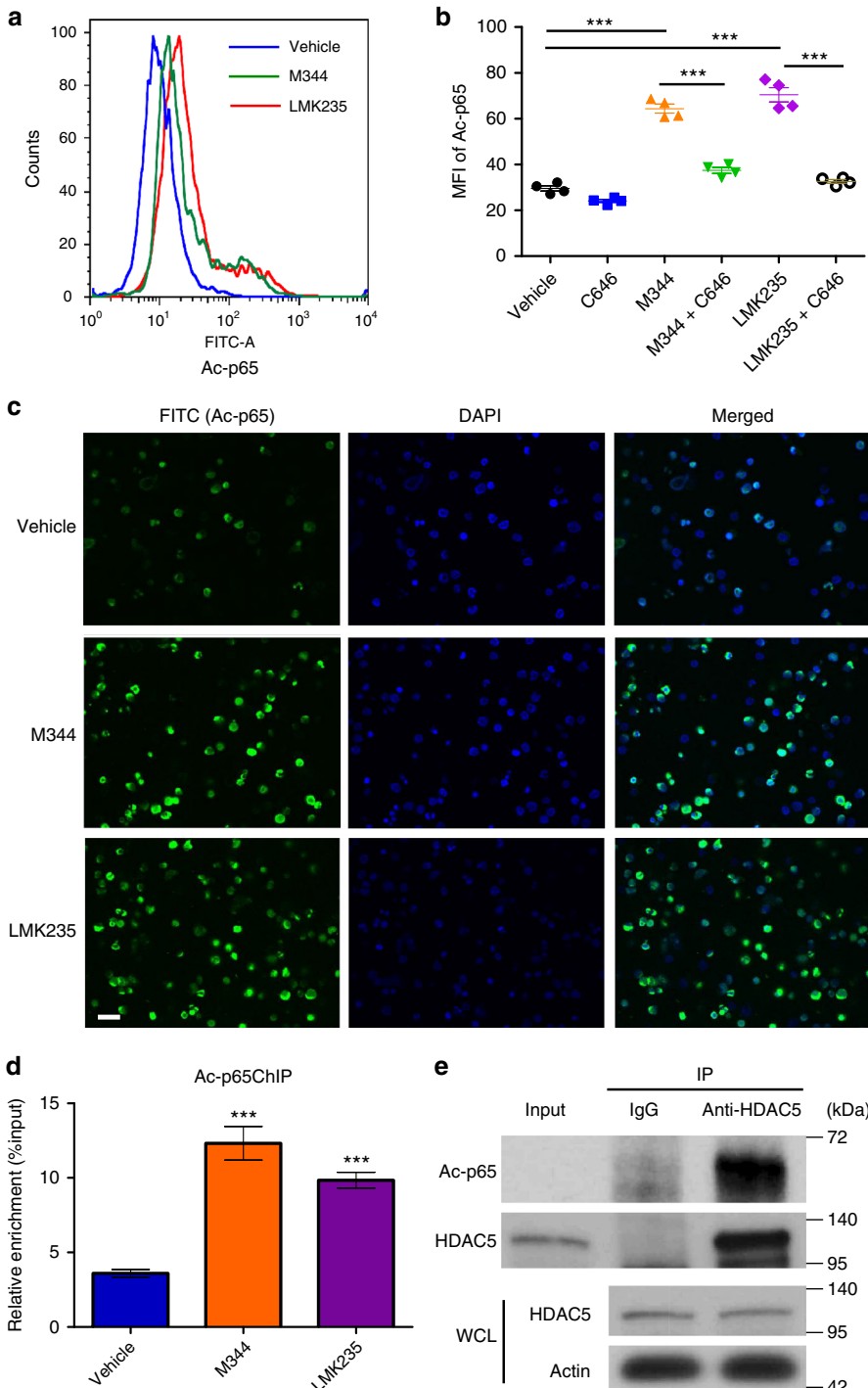

**Fig. 6** HDAC5 inhibition results in elevated levels of p65 acetylation. **a** Histogram of intracellular acetylated p65 (Ac-p65) levels of human CB CD34$^+$ cells treated with vehicle, M344, or HDAC5 inhibitor LMK235. Representative histogram from three independent experiments is shown. **b** Mean fluorescence intensity (MFI) of intracellular acetylated p65 (Ac-p65) levels in vehicle, C646 (50 μM), M344-, M344 + C646-, LMK235-, LMK235 + C646-treated human CB CD34$^+$ cells, as assessed by flow cytometry. Data pooled from two independent experiments are shown ($n = 4$, one-way ANOVA, ***$p < 0.001$). **c** Confocal imaging analysis of intracellular acetylated p65 (Ac-p65) levels of human CB CD34$^+$ cells treated with vehicle, M344, or LMK235. Green indicates intracellular acetylated p65 staining; DAPI (blue) labels the cell nucleus. Representative images from two independent experiments are shown. Scale bar: 20 μm. **d** Acetylated p65 levels on *CXCR4* promoter in vehicle-, M344-, or LMK235-treated human CB CD34$^+$ cells, as assessed by a ChIP assay. Data pooled from two independent experiments are shown ($n = 6$, one-way ANOVA, ***$p < 0.001$). **e** HDAC5 IP samples blotted with an anti-acetylated p65 antibody. HDAC5 and actin in the whole cell lysate (WCL) serve as loading controls. For all panels, data are shown as mean ± SEM

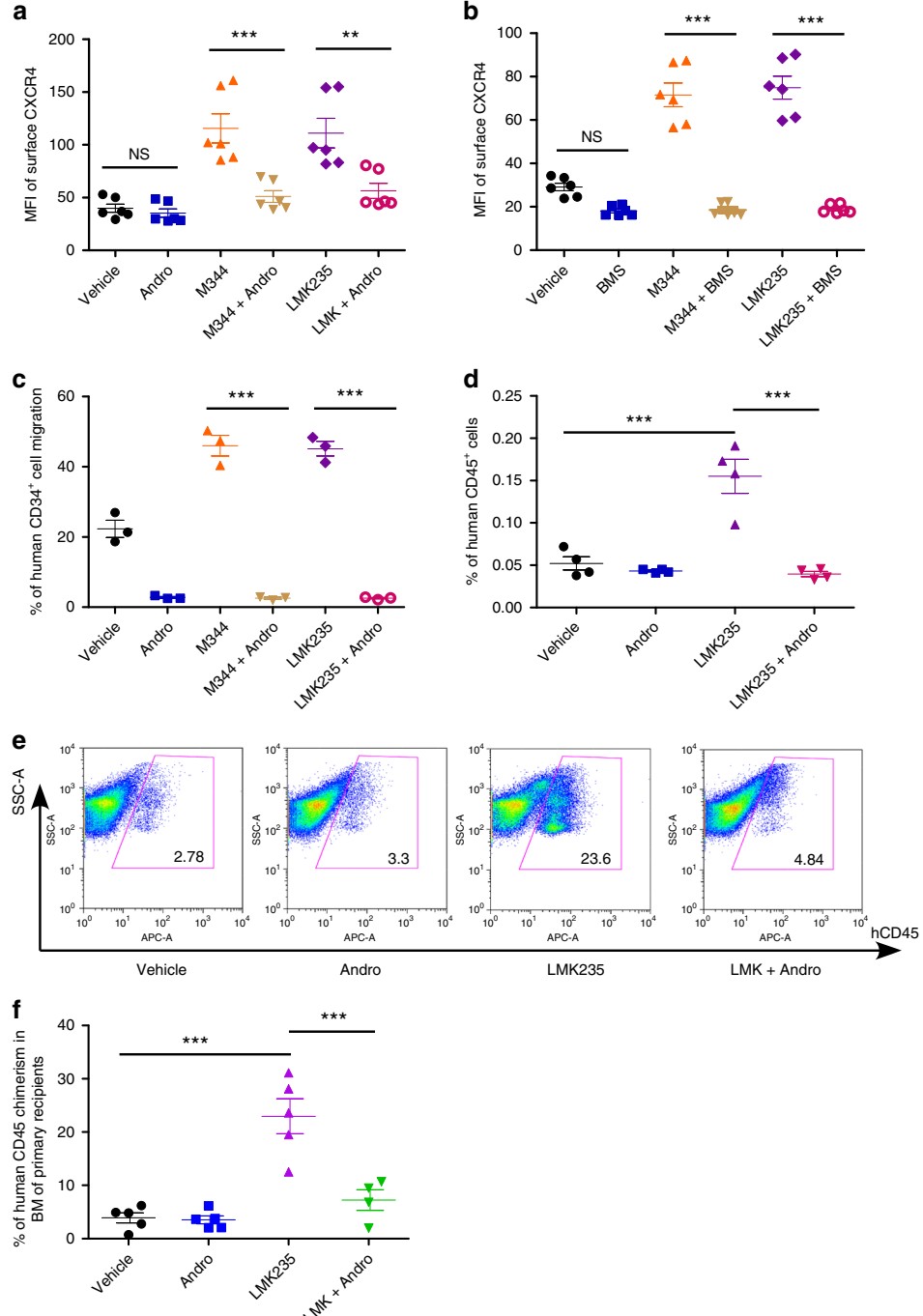

**Fig. 7** NF-κB signaling pathway is involved in HDAC5-mediated CB HSC homing. **a** Mean fluorescence intensity (MFI) of surface CXCR4 in vehicle, Andrographolide (Andro, 10 μM), M344-, M344 + Andro-, LMK235-, LMK235 + Andro-treated human CB CD34+ cells, as assessed by flow cytometry. Data pooled from three independent experiments are shown ($n = 6$, one-way ANOVA, **$p < 0.01$, ***$p < 0.001$, NS indicates $p > 0.05$). **b** Mean fluorescence intensity (MFI) of surface CXCR4 in vehicle, BMS345541 (BMS, 10 μM), M344-, M344 + BMS-, LMK235-, LMK235 + BMS-treated human CB CD34+ cells, as assessed by flow cytometry. Data pooled from three independent experiments are shown ($n = 6$, one-way ANOVA, ***$p < 0.001$, NS indicates $p > 0.05$). **c** The cells were cultured in the presence of vehicle, Andrographolide (Andro, 10 μM), M344, M344 + Andro-, LMK235, or LMK235 + Andro for 16 h and then allowed to migrate toward 50 ng/mL SDF-1 for 4 h. Data pooled from three independent experiments are shown ($n = 3$, one-way ANOVA, ***$p < 0.001$). **d** The percentage of human CD45+ cells in the BM of NSG mice 24 h after transplantation with 500,000 CB CD34+ cells that had been treated with vehicle, Andrographolide (Andro, 10 μM), LMK235, or LMK235 + Andro. Data pooled from four independent experiments are shown ($n = 4$ mice per group, one-way ANOVA, ***$p < 0.001$). **e** Representative flow cytometric analysis of human engraftment in the BM of NSG mice transplanted with human CB CD34+ cells treated with vehicle control, Andrographolide (Andro), LMK235, or LMK235 + Andro for 16 h. Human engraftment was assessed as the percentage of human CD45+ cells. **f** The percentage of human CD45+ cell chimerism in the BM of NSG mice after transplantation with 10,000 CB CD34+ cells that had been treated with vehicle control, Andrographolide (Andro), LMK235, or LMK235 + Andro for 16 h ($n = 4$-5 mice per group, one-way ANOVA, ***$p < 0.001$). For all panels, data are shown as dot plots (mean ± SEM)

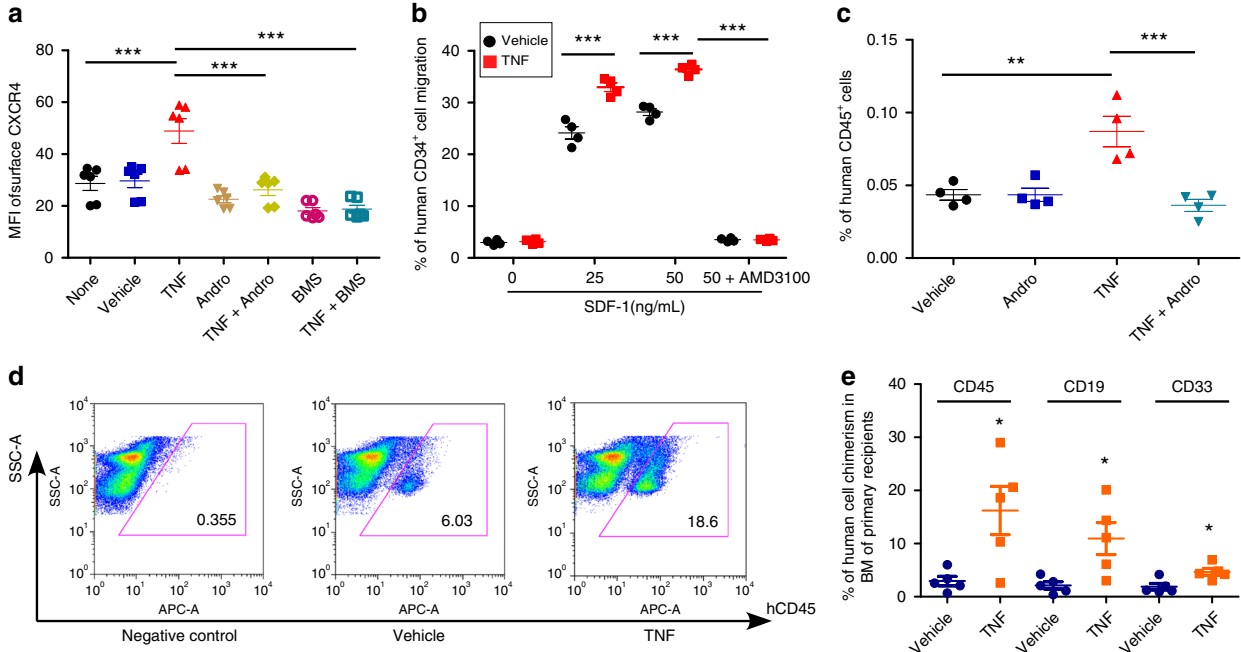

**Fig. 8** TNFα treatment results in significantly enhanced HSC homing and engraftment. **a** Mean fluorescence intensity (MFI) of surface CXCR4 in vehicle, TNFα (TNF, 100 ng/mL), Andrographolide (Andro)-, TNF + Andro-, BMS345541 (BMS)-, TNF + BMS-treated human CB CD34+ cells, as assessed by flow cytometry. None indicates the group without any treatment. Data pooled from three independent experiments are shown ($n = 6$, one-way ANOVA, ***$p < 0.001$). **b** The cells were cultured in the presence of vehicle or TNFα (TNF, 100 ng/mL) for 16 h and then allowed to migrate toward the indicated concentrations of SDF-1 for 4 h. Vehicle or TNF-treated human CB CD34+ cells migration in the presence of the CXCR4 antagonist, AMD3100 (5 μg/mL) were also shown. Data pooled from three independent experiments are shown ($n = 3$, one-way ANOVA, ***$p < 0.001$). **c** The percentage of human CD45+ cells in the BM of NSG mice 24 h after transplantation with 500,000 CB CD34+ cells that had been treated with vehicle, Andrographolide (Andro, 10 μM), TNFα (TNF, 100 ng/mL), or TNF + Andro. Data pooled from four independent experiments are shown ($n = 4$ mice per group, one-way ANOVA, **$p < 0.01$, ***$p < 0.001$). **d** Representative flow cytometric analysis of human engraftment in the BM of NSG mice transplanted with human CB CD34+ cells treated with vehicle control or TNFα (TNF, 100 ng/mL) for 16 h. Human engraftment was assessed as the percentage of human CD45+ cells. **e** The percentage of human CD45+ cells, B-cell (CD19+), and myeloid cell (CD33+) chimerism in the BM of NSG mice after transplantation with 10,000 CB CD34+ cells that had been treated with vehicle or TNFα (TNF, 100 ng/mL), $n = 5$ mice per group, $t$-test, *$p < 0.05$. For all panels, data are shown as dot plots (mean ± SEM)

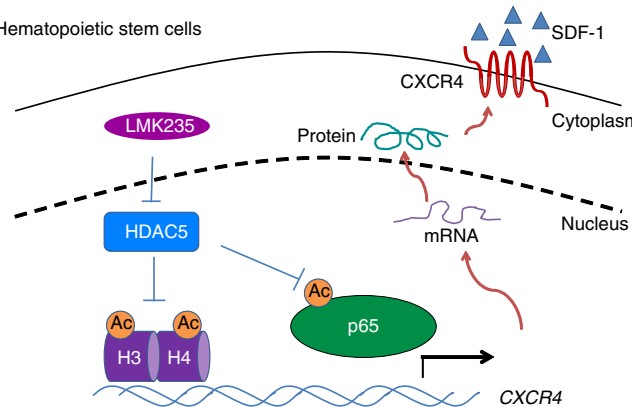

**Fig. 9** Model for the role of HDAC5 in regulating CXCR4 expression in human HSC/HPCs. HDAC5 inhibition by LMK235 results in enhanced acetylation of histone 3 on lysine 9 and histone 4 on lysine 16, as well as p65 acetylation. Acetylation of histone facilitates chromatin remodeling, whereas acetylation of p65 enhances its transcriptional activity. Acetylated p65 binds to CXCR4 promoter region and induces higher expression of CXCR4 and thus enhances HSC/HPC homing and engraftment

**Chemotaxis assay**. Chemotaxis assays were performed by using Costar 24-well transwell plates with 6.5 mm diameter inserts with 5.0 μm pores (Corning, Inc., Corning, NY, USA). 650 μL of pre-warmed IMDM medium (37 °C) that contained 0.5% bovine serum albumin (Sigma-Aldrich) and SDF-1 (0, 25, 50 ng/mL) was added to the bottom well. Cells were suspended at $1 \times 10^5$ cells/100 μL in IMDM

medium and loaded to the upper chamber of the transwell. Transwell plates were placed in a 37 °C incubator with 95% humidity and 5% CO$_2$ for 4 h. Percent migration was measured using flow cytometry with number of cells in the bottom chamber divided by number of cells placed in the upper chamber. To calculate percent migration of CB HSCs, phenotypic HSCs(CD34+CD38−CD45RA−CD90+ CD49f+ cells) was determined by surface staining and flow cytometry analysis. Human CB HSC chemotaxis was calculated as number of HSCs in the bottom chamber divided by number of HSCs loaded in the upper chamber. For AMD3100 administration, cells were treated with 5 μg/mL AMD3100 (239820, Sigma-Aldrich) for 30 min right before the chemotaxis assay.

**In vivo homing assay**. Homing of human CB CD34+ cells were evaluated in NSG mice. M344-, LMK235-, or vehicle control-treated CB CD34+ cells (500,000 for each mouse) were intravenously injected into sublethally irradiated (350 cGy) NSG mice. After 24 h, these recipient mice were sacrificed, BM and spleen cells from each mouse were collected. Cells were stained with anti-human CD45 antibody, then resuspended in 1% formaldehyde buffer. Flow cytometry analysis was performed to determine the percentage of human CD45+ cells. BM and spleen cells from non-transplanted NSG mice were served as the negative control.

**Limiting dilution analysis**. Limiting dilution analysis was performed to calculate the frequency of human SRCs in different experimental groups[16,49]. Briefly, increasing doses of M344-, LMK235-, or vehicle control-treated CB CD34+ cells (500, 2500, or 10,000 cells) were intravenously transplanted into sublethally irradiated (350 cGy) NSG mice. Sixteen weeks after transplantation, BM cells were collected and human CD45+ cell, B-cell, and myeloid cell chimerism was evaluated by immunostaining and flow cytometry analysis. For the secondary engraftment and assessment of self-renewal activity, $5 \times 10^6$ BM cells from the primary NSG recipient mice were injected into secondary NSG recipients. The HSC frequency was calculated using L-Calc software (Stem Cell Technologies Inc, Vancouver, BC, Canada) and plotted using ELDA software (bioinf.wehi.edu.au/software/elda/).

**ChIP assay**. ChIP was conducted using an EZ-ChIP kit following the manufacturer's instructions (17-371, EMD Millipore, Billerica, MA, USA)[16]. In brief, vehicle control-, M344-, or LMK235-treated CB CD34[+] cells were fixed in 1% formaldehyde for 10 min, and then lysed, sonicated, and incubated with specific antibodies, followed by precipitation using protein G agarose beads. Then enriched chromatin DNA was isolated and subjected to quantitative real-time PCR analysis. Data are presented as the percentage of input. The following primers were used for CXCR4 ChIP qPCR analysis: 5′- TTCCAGTGGCTGCATGTGTC-3′ and 5′-CAGACAATGTAACTCGCTCC-3′. The antibodies used for ChIP assay were as follows: anti-p65 acetyl K310 (ab19870, Abcam), anti-H3K9ac (06-942, Millipore, Kankakee, IL, USA), and anti-H4K16ac (ab109463, Abcam).

**Immunoprecipitation and immunoblotting**. Immunoprecipitation assay was performed to verify the endogenous interaction between HDAC5 and acetylated p65[16]. 293T Cells (ATCC CRL-3216) were lysed using immunoprecipitation buffer containing protease inhibitor cocktail. Cell lysates were then centrifuged at 4 °C, 13,000 × g for 15 min, and the supernatants were incubated with anti-HDAC5 antibody (Cell Signaling Technology, Beverly, MA, USA, D1J7V, 20458) or IgG control in 1:100 dilution, followed by incubation withprotein A agarose beads (Cell Signaling Technology). Immunoprecipitates were washed and subjected to immunoblotting analysis using anti-p65 acetyl K310 (Abcam) and anti-actin (Sigma, A3853). Uncropped images of western blottings are shown in Supplementary Figure 7.

**RNA extraction and real-time PCR**. RNA was extracted using the RNeasy Mini Kit following the manufacturer's protocol (74104, QIAGEN, Valencia, CA, USA). Total RNA was reverse transcribed using Superscript III First-Strand Synthesis System (Thermo Fisher, Waltham, MA, USA). Quantitative real-time PCR reactions were performed using the QuantiTect SYBR Green PCR Kit (204143, QIAGEN) and a Stratagene Mx3000p PCR System. GAPDH mRNA levels were used as the internal control. mRNA expression levels were normalized to GAPDH mRNA levels and shown as arbitrary units relative to vehicle control (set as 1). The primer sequences for CXCR4 were: 5′- TCTATGTTGG CGTCTGGATCC-3′ and 5′-CTTGGAGTGTGACAGCTTGG-3′. GAPDH primer sequences were: 5′-TTCGTCATGGGTGTGAACCA-3′ and 5′- TGGCAGT GATGGCATGGACT-3′.

**RNA-seq and analysis**. CB CD34[+] cells were pretreated with vehicle control or LMK235 for 16 h. RNA was extracted using the RNeasy Mini Kit (QIAGEN). RNA-seq was conducted using Illumina Nextseq 500 in the Center for Medical Genomics at Indiana University School of Medicine. For RNA-seq analysis, we used FastQC to examine the quality of RNA-seq reads then mapped all sequences past the quality control to the human genome (UCSC hg19) by STAR RNA-seq aligner[50]. Uniquely mapped sequencing reads were assigned to genes according to the refGene from UCSC (hg19) with the package of Subread, featureCounts[51]. The gene expression was normalized by the method of trimmed mean of M values. EdgeR[52] was adopted for differential expression analysis by paired comparison between each paired replicates of treatment and control. P-value of each gene was adjusted by multiple test correction via false discovery rate (FDR) estimation. Differentially expressed gene (DEG) was determined if its FDR-adjusted p-value was less than 0.01, and the fold change was greater than 1.7 (upregulated) or less than 0.588 (downregulated). Only DEG whose average count-per-million for wild-type samples is larger than 2 was took into account for biological functional analysis by using DAVID functional annotation analysis (http://david.abcc.ncifcrf.gov/home.jsp) v6.8)[53,54].

**Statistical analysis**. Statistical analysis was conducted using Microsoft Excel and GraphPad Prism. Two-tailed Student's t-test was used for statistical analysis between two groups. One-way analysis of variance with Tukey's multiple comparison test was performed to compare the difference among more than two groups where indicated. P values are designated as *p < 0.05, **p < 0.01, ***p < 0.001. Error bars represent SEM unless otherwise stated.

**Data availability**. RNA-seq data have been deposited in the GEO database under accession number GSE110736. The authors declare that all data supporting the findings of this study are available within the article and its supplementary information files or from the corresponding author upon reasonable request.

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

## Acknowledgements

This work was supported by Grants from the NIH to HEB: (R01 HL112669, R01 HL056416, R35 HL139599, U54 DK106846). RNA sequencing was carried out in the Center for Medical Genomics at Indiana University School of Medicine, which is partially supported by the Indiana Genomic Initiative at Indiana University (INGEN); INGEN is supported in part by the Lilly Endowment, Inc. The bioinformatics analysis was done by Collaborative Core for Cancer Bioinformatics (C3B) shared by Indiana University Simon Cancer Center (P30CA082709) and Purdue University Center for Cancer Research (P30CA023168) with support from the Walther Cancer Foundation. We also want to thank Broxmeyer laboratory personnel for helpful assistance and discussions, and the In Vivo Therapeutics Core and the Flow Cytometry Facility of the Indiana University School of Medicine, funded in part by U54 DK106846, for assistance.

## Author contributions

X.H. and H.E.B. conceived the project and designed the experiments. X.H. and B.G. performed the experiments and analyzed the data. S.L. and J.W. performed bioinformatics analysis on RNA sequence data. X.H., B.G., and H.E.B. wrote and edited the manuscript. It is noteworthy that X.H. and B.G. contributed equally to this work.
