## [Peer Review File · Nature Communications]

Reviewers' comments:

Reviewer #1 (Remarks to the Author):

In this study, the authors screened HDAC inhibitors and HDACs that enhance human HSC function. They found that specific HDAC5 inhibition highly upregulates CXCR4 surface expression in human CB HSCs and HPCs, resulting in enhanced SDF-1/CXCR4 mediated chemotaxis and increased homing to the bone marrow, with elevated SCID-repopulating cell (SRC) frequency and enhanced long term engraftment in NSG mice. HDAC5 inhibition increased acetylated p65 levels in the nucleus, which is important for CXCR4 transcription, unveiling the role of NF- κ B signaling in HDAC5-mediated CXCR4 upregulation and enhanced HSC homing. These findings suggest that neutralizing negative epigenetic regulation of HSC homing and engraftment by HDAC5 and NF- κ B signaling would be a new strategy to enhance HSC transplantation.

This study is well designed and the data are solid. The findings are new and have impacts on the epigenetic manipulation of HSCs for transplantation therapy. To improve the manuscript, I have several points to be addressed.

1. The authors failed to describe the current knowledge of HDAC5 function in the text. Please give the readers more information on HDAC5.
2. CXCR4 expression at the mRNA level should be validated in the treatment with M344, shHDAC5, LMK235, and TNF α . Is its expression really regulated at the transcriptional level?
3. CXCR5 would not be the only targets of HDAC5. To know the other targets, it would be very informative to do RNA-seq of CD34+ cells treated with HDAC inhibitor or shHDAC5 compared with non-treated CD34+ cells.
4. Does HDAC5 physically interact with p65 in co-ip experiment?
5. Please show the expression profiles of HDAC5 in HSPCs. Is it preferentially expressed in immature cell fractions?
6. The labeling "GFP" should be "HDAC5" in Figure 3g.

Reviewer #2 (Remarks to the Author):

This is an interesting and translationally significant study that describes the impact of HDAC inhibition on human CB HSC homing and long-term engraftment in NSG mice and describes a mechanism of action in which HDAC inhibition augments HSC homing via induction of CXCR4. I have some concerns about the data presentation and conclusions, which if addressed, will improve the manuscript.

Figure 1. The magnitude of difference between the M344 – treatment group and the vehicle control group in the secondary mice (1m) is less than the quantitation of the SRC frequency would suggest. If there are 4.3 - fold more SRCs in the M344 – treated CB cells, then the difference in secondary transplanted NSGs should, in principle, be greater than shown. Or is the conclusion that M344 increases the frequency of short term repopulating HSCs capable of engraftment in primary transplanted NSGs, but not sustained in secondary mice?

The NSG studies in Figure 1 are interesting and central to the conclusions of the paper. It would be beneficial if more than n=5 mice per group were tested in these studies, given their importance. Were these studies repeated more than once?

Figure 2. M344, utilized in Figure 1, has been shown in other studies to have inhibitory activity against HDAC1 (Huber et al. J Biol Chem 2011), whereas the analysis in Figure 2a suggests that only HDAC5 inhibition yields an increase in surface CXCR4. Does this reflect the lack of specificity of M344 or how do the authors connect their findings in Figure 1 with the observation in Figure 2a?

LMK235 is described as selective for HDAC4 and HDAC5 when used in the nanomolar range (Marek et al. J Med Chem 2013). Since micromolar dose range of LMK235 is used here (Methods), have the authors confirmed the specificity of this compound for HDAC4 and HDAC5 or is it possible that at the higher molar concentrations, this HDAC inhibitor is no longer specific for these 2 HDACs alone?

Figure 2f,g. While the differences in CD34 cell migration and HSC migration represented in these figures meets statistical significance, the numbers of replicates (n=3) suggests these experiments were performed only one; it would be better if such experiments were repeated at least once.

Figure 2j. The total CD45 donor cell engraftment in the LMK235 treatment group is nearly 6%, yet the percentage donor – derived CD19+ cells and CD33+ cells appear to be less than 2% each. Are there unmeasured T cells that are contributing to the total human cell engraftment in the NSG mice at this time point?

Figure 2m. If there is a 6-fold increase in SRC frequency in the LMK235 treatment graft, why is the donor human cell chimerism only 2-3 fold increased in the secondary mice represented in Figure 2m? Is LMK235 amplifying a short term human HSC with some exhaustion detected in secondary transplanted mice?

Figure 4. The mechanistic studies represented here are well done and convincing. It has been suggested that andrographolide has other molecular targets beside NFkB, including Nrf2 (Wong et al. J Neuroinflamm 2016). Please comment.

The authors show effects on HSC homing in Figure 4g, but it would be important to know whether the inhibition of NFkB signaling abrogates the longer term HSC engraftment that is enhanced by treatment with M344 or LMK235. Have the authors completed such studies to include in Figure 4?

The data presented here demonstrate that treatment of human CB cells with either M344 or LMK235 for 16 hours in vitro increases the homing of CB cells in NSG mice and this produces increased long term human hematopoietic cell engraftment through 4 months. Previously, Ron Hoffman's laboratory showed that ex vivo culture of human CB cells with valproic acid or scriptaid (SCR), both HDAC inhibitors, caused the expansion of human phenotypic CB HSCs in short term culture (Chaurasia et al. JCI 2014). Similar to the observations here, Chaurasia et al. demonstrated that HDAC inhibitor treatment increased the homing of human CB HSCs in NSG mice. Additionally, one of the compounds, SCR, was shown to substantially decreased HDAC5 protein expression levels, the target of the current study. The authors need to address their current findings in the context of these prior results as to whether the prior studies from the Hoffman lab predicted the observations here. Conversely, the authors should highlight more clearly how their study adds substantially to the prior conceptual advance.

HDAC inhibition has been shown in prior studies to alter CXCR4 expression (Kim et al. Cell Immunol 2013; Tsai L et al. Neuropsychopharm 2010; Matalon et al. J AIDS 2010) in non-HSC cell types. The authors should cite or note this prior art relating to this mechanism of HDAC inhibitor action.

Reviewer #3 (Remarks to the Author):

Huang X et al. Neutralizing negative epigenetic regulation by HDAC5 enhances human haematopoietic stem cell homing and engraftment.

The authors report that a) NFkB is a transcriptional factor on a previously unrecognized target gene, CXCR4. This is interesting because other CXCR genes had been reported to be downstream targets of NFkB (CXCR1 and CXCR2) but not the HSC relevant CXCR4; b) HDAC5 inhibition upregulates CXCR4 transcription and functional surface expression (~3-fold); c) HDAC5 inhibition increases acetylated p65 levels in the nucleus and HSC homing/engraftment; d) inhibition of NFkB signaling (directly with andrographolide or of IKK by BMS345541) suppressed both HDAC5-mediated CXCR4 upregulation and enhanced HSC homing; e) activation of NFkB signaling via TNFa results in increased functional CXCR4 surface expression and enhanced HSC homing.

The manuscript is well written and the experiments and conclusions are well conceived and drawn. Limiting dilution experiments allowed the quantification of SRCs of manipulated human CD34+ cells. The manuscript uses abundant pharmacological tools as well as one shRNAs for each HDAC I/IIA(1-8). No information is provided on the effect of shRNAs to HDAC IIB (except HDAC6), III, and IV subtypes.

In addition to NFkB and IKK inhibitors, the pharmacological inhibitors used are:

1. M344, a HDAC inhibitor with subtype selectivity for HDAC6 over HDAC1. M344 inhibits HDAC (IC50 = 100 nM) and also inhibits hyperacetylation of histone H4.
2. LMK-235 is a HDACi that exhibits HDAC isoform selectivity, with a preference for HDAC4 and HDAC5.

Other HDACi with limited specificity like vorinostat, thicostatin A and belinostat are used and find similar effects on CXCR4 expression.

3. AMD3100, a CXCR4 inhibitor.

There are only a few concerns that would require attention:

1. Major: a) The paper would benefit from information on self-renewing HSC homing/engraftment by providing information on secondary recipient engraftment of homed HSC in primary recipients. b) characterization of CXCR4 expression in further refined CD34+ cell subpopulations accounting for self-renewing HSC (CD34+/CD38-/CD90+/CD45RA-/Rh123low/CD49f+) or at the very least CD34+/CD38-/CD90+ cells. As presented, the information is focused on non-HSC but mostly on MPP or ST-HSC.

2. Minor: Figure 3: a) Figure 3e would benefit from a histogram plot including the effect of M344. b) Information on Ac-p65 levels after combinations of M344+C646 and LMK235+C646 to Figure 3 would also be warranted.

We would like to thank all the reviewers and the editor for their valuable suggestions for improvement of our manuscript. We have conducted further experiments and reorganized the manuscript in response to the reviewers' comments. We believe that the manuscript is substantially improved as a result.

Point-by-point responses to reviewers' comments:

Reviewer #1 (Remarks to the Author):

In this study, the authors screened HDAC inhibitors and HDACs that enhance human HSC function. They found that specific HDAC5 inhibition highly upregulates CXCR4 surface expression in human CB HSCs and HPCs, resulting in enhanced SDF-1/CXCR4 mediated chemotaxis and increased homing to the bone marrow, with elevated SCID-repopulating cell (SRC) frequency and enhanced long term engraftment in NSG mice. HDAC5 inhibition increased acetylated p65 levels in the nucleus, which is important for CXCR4 transcription, unveiling the role of NF- κ B signaling in HDAC5-mediated CXCR4 upregulation and enhanced HSC homing. These findings suggest that neutralizing negative epigenetic regulation of HSC homing and engraftment by HDAC5 and NF- κ B signaling would be a new strategy to enhance HSC transplantation.

This study is well designed and the data are solid. The findings are new and have impacts on the epigenetic manipulation of HSCs for transplantation therapy. To improve the manuscript, I have several points to be addressed.

Response: We thank this reviewer for carefully reviewing the manuscript and the supportive statement with the constructive comments.

1. The authors failed to describe the current knowledge of HDAC5 function in the text. Please give the readers more information on HDAC5.

Response: We thank the reviewer for raising this point. This comment has now been addressed in the revised manuscript. We have added information of HDAC5 in the introduction and discussion (see the third paragraph of Introduction and the third paragraph of Discussion).

2. CXCR4 expression at the mRNA level should be validated in the treatment with M344, shHDAC5, LMK235, and TNFa. Is its expression really regulated at the transcriptional level?

Response: We thank the reviewer for this valuable suggestion. As requested by the reviewer, we have added the results of RT-PCR for shHDAC5 and TNF α in the revised manuscript. Significant upregulation of CXCR4 mRNA was noticed in the treatment with M344, LMK235, shHDAC5 and TNF α (Fig. 3a, b and Supplementary Fig. 6a). Moreover, RNA-seq analysis showed that CXCR4 was strongly enriched in LMK235 treated CD34⁺ cells (Fig. 3f). Increased histone acetylation levels were also confirmed at the CXCR4 promoter region (Fig. 3c, d). Taken together, we believe that CXCR4 expression is tightly regulated at the transcriptional level.

3. CXCR5 would not be the only targets of HDAC5. To know the other targets, it would be very informative to do RNA-seq of CD34⁺ cells treated with HDAC inhibitor or shHDAC5 compared with non-treated CD34⁺ cells.

Response: We thank the reviewer for this insightful comment. We agree that CXCR4 would not be the only target of HDAC5. As suggested by the reviewer, we have performed RNA sequencing analysis on LMK235 and vehicle control treated CD34⁺ cells (Fig. 3f, g). As expected, CXCR4 was strongly enriched in LMK235 treated CD34⁺ cells (Fig. 3f). Consistent with the reports showing that PGE2 promotes HSC homing, we also identified that PTGER4, a PGE2 receptor, has a strong enrichment in LMK235 treated CB CD34⁺ cells. Gene ontology (GO) analysis revealed an upregulation of gene sets linked with cell locomotion, cell motility, cell migration, cell adhesion, and cytoskeleton (Fig. 3g). We believe that the transcriptomic data provide additional evidence supporting that HDAC5 modulates HSC homing through transcriptional regulation.

4. Does HDAC5 physically interact with p65 in co-ip experiment?

Response: We thank the reviewer for this constructive comment. As requested by the reviewer, we performed immunoprecipitation experiments to detect endogenous HDAC5 and p65 interaction. We found that HDAC5 immunoprecipitated with acetylated p65 (Fig. 4e), suggesting HDAC5 can physically interact with acetylated p65 to remove acetylation from p65. Consistent with this, HDAC5 inhibition resulted in elevated levels of p65 acetylation (Fig. 4a-c).

5. Please show the expression profiles of HDAC5 in HSPCs. Is it preferentially expressed in immature cell fractions?

Response: We thank the reviewer for this constructive comment. As requested by the reviewer, we analyzed HDAC5 expression in HSCs, MPPs, CD34⁺CD38⁻ HSPCs and CD34⁻ cells (Supplementary Fig. 3g, h). We found that HDAC5 was highly expressed in HSCs suggesting a potential role of HDAC5 in HSCs.

6. The labeling “GFP” should be “HDAC5” in Figure 3g.

Response: This comment has now been addressed in the revised manuscript. We have added “FITC (Ac-p65)” to make this more clearly (Fig. 4c).

Reviewer #2 (Remarks to the Author):

This is an interesting and translationally significant study that describes the impact of HDAC inhibition on human CB HSC homing and long-term engraftment in NSG mice and describes a mechanism of action in which HDAC inhibition augments HSC homing via induction of CXCR4. I have some concerns about the data presentation and conclusions, which if addressed, will improve the manuscript.

Response: We thank this reviewer for carefully reviewing the manuscript and the constructive comments.

Figure 1. The magnitude of difference between the M344 – treatment group and the vehicle control group in the secondary mice (1m) is less than the quantitation of the SRC frequency would suggest. If there are 4.3 - fold more SRCs in the M344 – treated CB cells, then the difference in secondary transplanted NSGs should, in principle, be greater than shown. Or is the conclusion that M344 increases the frequency of short term repopulating HSCs capable of engraftment in primary transplanted NSGs, but not sustained in secondary mice?

Response: We thank this reviewer for carefully reading our manuscript and raising this comment. The SRC frequency was calculated by poisson statistical analysis using limiting dilution assay. This methodology usually results in bigger magnitude of differences in SRCs than human CD45 percentage chimerism. For example, SR1 treatment resulted in 17-fold increase in SRCs while only ~2 fold increase in average human CD45 chimerism (Boitano et al., Science, 2010); UM171 treatment had a 13-fold increase in SRCs while only 1.1 fold increase in average human CD45 chimerism (Fares et al., Science, 2014); VPA treatment had a 36 fold increase in SRCs while only ~ 2 fold increase in average human CD45 chimerism in secondary transplanted NSGs (Chaurasia et al., J Clin Invest, 2014).

In our study, the magnitude of difference in human CD45 chimerism for primary NSG mice is 2.34 fold (Fig. 1j), while the magnitude is 2.27 fold for secondary NSG mice (Fig. 1m). Both of the increase in primary and secondary NSG mice were statistically significant, suggesting that the enhancement of human cell engraftment by M344 treatment is sustained in serial transplantations. Our results indicate that M344 treatment resulted in increased engraftment of long term CB HSCs in NSG mice.

The NSG studies in Figure 1 are interesting and central to the conclusions of the paper. It would be beneficial if more than n=5 mice per group were tested in these studies, given their importance. Were these studies repeated more than once?

Response: As requested by the reviewer, we performed additional transplantation studies to increase the number of mice in Fig. 1h,j and Fig. 2g, j. All of these studies were reproducible and repeated more than once, which is now stated in our figure legends.

Figure 2. M344, utilized in Figure 1, has been shown in other studies to have inhibitory activity against HDAC1 (Huber et al. J Biol Chem 2011), whereas the analysis in Figure 2a suggests that only HDAC5 inhibition yields an increase in surface CXCR4. Does this reflect the lack of specificity of M344 or how do the authors connect their findings in Figure 1 with the observation in Figure 2a?

Response: M344 is pan-HDAC inhibitor which targets all the HDACs without specificity. We tested three other pan-HDAC inhibitors (Vorinostat, Trichostatin A and Belinostat) and found that their treatment also resulted in upregulation of surface CXCR4 expression. Different HDACs have different target preferences according to their substrate selectivity. To identify which HDAC is responsible for the regulation of CXCR4 expression, we utilized shRNA knockdown (Fig 2a) and various selective HDAC inhibitors (Supplementary Fig. 3b) to examine their effects on human CB CD34⁺ cells. We found that only HDAC5 shRNA and HDAC5 inhibitor upregulated surface CXCR4 expression. The data in Fig. 2 is one step further relative to Fig. 1, and demonstrates the application of a more specific HDAC inhibitor to enhance CB HSC homing, without modifying expression of other unnecessary target genes.

LMK235 is described as selective for HDAC4 and HDAC5 when used in the nanomolar range (Marek et al. J Med Chem 2013). Since micromolar dose range of LMK235 is used here(Methods), have the authors confirmed the specificity of this compound for HDAC4 and HDAC5 or is it possible that at the higher molar concentrations, this HDAC inhibitor is no longer specific for these 2 HDACs alone?

Response: We thank the reviewer for bringing up this point. We agree that in the original publication, LMK235 showed the highest activity at HDAC5 in the nanomolar range (Marek et al. J Med Chem 2013). However, their assay is an in vitro activity assay. To target the intracellular HDAC5 when treating cells, we believe that higher concentrations of LMK235 are needed. Also different cell types will be different in the capability to take up the compound. We performed titration on LMK235 for human CB CD34⁺ cells and determined that 1µM worked best (Supplementary Fig 3a). So we used 1µM to treat human CB CD34⁺ cells for in vivo and mechanism studies.

It is technically difficult to measure HDAC activity inside human CB CD34⁺ cells. So we used shRNA knockdown and showed that only HDAC5 knockdown upregulated CXCR4 proteins and mRNA levels (Fig. 2a and Fig. 3b). We also found that other selective HDAC inhibitor did not increase surface CXCR4 expression

(Supplementary Fig 3b). Taken together, we believe that HDAC5 is the key component in regulating CXCR4 expression and homing of human CB CD34⁺ cells.

Figure 2f,g. While the differences in CD34 cell migration and HSC migration represented in these figures meets statistical significance, the numbers of replicates (n=3) suggests these experiments were performed only one; it would be better if such experiments were repeated at least once.

Response: We apologize for any confusion in our previous description. The chemotaxis experiments were repeated three times using three different cord blood samples. We have revised the figure legend to make it more clear. Now it is stated “Data pooled from three independent experiments are shown (each dot represents an independent chemotaxis, n=3 CB samples, one-way ANOVA, ***p<0.001).”

Figure 2j. The total CD45 donor cell engraftment in the LMK235 treatment group is nearly 6%, yet the percentage donor – derived CD19⁺ cells and CD33⁺ cells appear to be less than 2% each. Are there unmeasured T cells that are contributing to the total human cell engraftment in the NSG mice at this time point?

Response: We thank the reviewer for raising this point. We have performed additional transplantation experiments to verify our results. In the revised manuscript, we have 10 mice for each group and we found the CD19⁺ and CD33⁺ cells are the main population inside CD45⁺ cells (Fig. 2j, CD45 average chimerism is 14.2%, CD19 is 8.08% and CD33 is 4.62%). We also measured T cell chimerism using CD3 antibody, however, the levels of CD3 chimerism is very low (less than 1%), so we did not show the data of CD3. Low T cell chimerism has also been described for human cord blood transplantations in our previous publication (Guo et al., Nat Med, 2016) and in publications from other groups in this field (Fares et al., Science, 2014; Boitano et al., Science, 2010).

Figure 2m. If there is a 6-fold increase in SRC frequency in the LMK235 treatment graft, why is the donor human cell chimerism only 2-3 fold increased in the secondary mice represented in Figure 2m? Is LMK235 amplifying a short term human HSC with some exhaustion detected in secondary transplanted mice?

Response: Please also see our response to the first point. The SRC frequency was calculated by poisson statistical analysis using limiting dilution assay. This methodology usually results in bigger magnitude of difference in SRCs than human CD45 percentage chimerism, which is shown by our previous work and publications from other groups (Guo et al., Nat Med, 2016; Fares et al., Science, 2014; Boitano et al., Science, 2010).

Figure 4. The mechanistic studies represented here are well done and convincing. It

has been suggested that andrographolide has other molecular targets beside NFκB, including Nrf2 (Wong et al. J Neuroinflamm 2016). Please comment.

Response: We thank the reviewer for the supportive comments regarding our mechanistic studies. We have used the other two NF-κB inhibitors (BMS345541 and PDTC) and confirmed that treatment of these NF-κB inhibitors suppressed the effects of M344 and LMK235 on human CB CD34⁺ cells (Fig. 5b and Supplementary Fig. 5a, b). So we believe that the effect of Andrographolide on CXCR4 expression is mainly through targeting NF-κB signaling pathway.

The authors show effects on HSC homing in Figure 4g, but it would be important to know whether the inhibition of NFκB signaling abrogates the longer term HSC engraftment that is enhanced by treatment with M344 or LMK235. Have the authors completed such studies to include in Figure 4?

Response: This comment has now been addressed in the revised manuscript. As requested by the reviewer, we performed additional engraftment assays with vehicle, Andrographolide, LMK235 and LMK235+Andrographolide treated CB CD34⁺ cells. We confirmed that Andrographolide also suppressed LMK235-mediated increases of human cell chimerism in primary recipient mice (Fig. 5e, f).

The data presented here demonstrate that treatment of human CB cells with either M344 or LMK235 for 16 hours in vitro increases the homing of CB cells in NSG mice and this produces increased long term human hematopoietic cell engraftment through 4 months. Previously, Ron Hoffman's laboratory showed that ex vivo culture of human CB cells with valproic acid or scriptaid (SCR), both HDAC inhibitors, caused the expansion of human phenotypic CB HSCs in short term culture (Chaurasia et al. JCI 2014). Similar to the observations here, Chaurasia et al. demonstrated that HDAC inhibitor treatment increased the homing of human CB HSCs in NSG mice. Additionally, one of the compounds, SCR, was shown to substantially decreased HDAC5 protein expression levels, the target of the current study. The authors need to address their current findings in the context of these prior results as to whether the prior studies from the Hoffman lab predicted the observations here. Conversely, the authors should highlight more clearly how their study adds substantially to the prior conceptual advance.

Response: We thank the reviewer for bringing up this comment and we agree that the work by Chaurasia et al. is an important and conceptual advance. They defined the concept that epigenetic reprogramming promoted HSC expansion and engraftment. We have cited their paper in the revised manuscript.

There are several advancements regarding our current study compared with their work. First, we used a different methodology. We treated CD34⁺ cells for only 16 hours and focused on HSC homing, while Chaurasia et al. treated CD34⁺ cells for 7 days and

mainly focused on HSC expansion. Reduced treatment time will allow for easier translation to clinical application.

Second, Chaurasia et al. did not identify which HDAC is involved. Although scriptaid (SCR) was shown to substantially decrease HDAC5 protein expression, it also largely decreased HDAC1, HDAC2, HDAC3 and HDAC4 protein expression. So SCR is a pan-HDAC inhibitor, which is also supported by the literature (Su et al., Cancer Res, 2000). In our study we used shRNA and HDAC5 inhibitor to show that HDAC5 is specifically involved in HSC homing process.

Third, we have also demonstrated for the first time that NF- κ B signaling is involved in HSC homing to the BM microenvironment. Inhibition of NF- κ B signaling suppressed the effects of HDAC5 inhibitor on CXCR4 upregulation and enhanced HSC homing, while activation of NF- κ B signaling by TNF α promoted CXCR4 expression and HSC homing.

HDAC inhibition has been shown in prior studies to alter CXCR4 expression (Kim et al. Cell Immunol 2013; Tsai L et al. Neuropsychopharm 2010; Matalon et al. J AIDS 2010) in non-HSC cell types. The authors should cite or note this prior art relating to this mechanism of HDAC inhibitor action.

Response: We thank the reviewer for providing this information. We have added discussion about this and cited these publications. Now it is stated “HDAC inhibition has been shown to alter CXCR4 expression in various cell types, and it seems the regulation could be either positive or negative depending on cell context. HDAC inhibition decreased surface CXCR4 expression in dendritic cells, CD4 T cells and monocytes^{44,45}, whereas inhibition of HDAC promoted surface CXCR4 expression in mesenchymal stem cells⁴⁶ and HSCs (this study). It would be interesting to explore what kind of cell type specific factors cooperated with HDAC to regulate CXCR4 expression in various types of cells.”

Reviewer #3 (Remarks to the Author):

Huang X et al. Neutralizing negative epigenetic regulation by HDAC5 enhances human haematopoietic stem cell homing and engraftment.

The authors report that a) NF κ B is a transcriptional factor on a previously unrecognized target gene, CXCR4. This is interesting because other CXCR genes had been reported to be downstream targets of NF κ B (CXCR1 and CXCR2) but not the HSC relevant CXCR4; b)HDAC5 inhibition upregulates CXCR4 transcription and functional surface expression (~3-fold); c) HDAC5 inhibition increases acetylated p65 levels in the nucleus and HSC homing/engraftment; d) inhibition of NF κ B signaling (directly with andrographolide or of IKK by BMS345541)suppressed both HDAC5-mediated CXCR4 upregulation and enhanced HSC homing; e) activation of NF κ B signaling via TNF α results in increased functional CXCR4 surface expression and enhanced HSC homing.

The manuscript is well written and the experiments and conclusions are well conceived and drawn. Limiting dilution experiments allowed the quantification of SRCs of manipulated human CD34+ cells.

The manuscript uses abundant pharmacological tools as well as one shRNAs for each HDAC I/IIA(1-8). No information is provided on the effect of shRNAs to HDAC IIB (except HDAC6), III, and IV subtypes.

In addition to NFkB and IKK inhibitors, the pharmacological inhibitors used are:

- 1. M344, a HDAC inhibitor with subtype selectivity for HDAC6 over HDAC1. M344 inhibits HDAC (IC50 = 100 nM) and also inhibits hyperacetylation of histone H4.*
- 2. LMK-235 is a HDACi that exhibits HDAC isoform selectivity, with a preference for HDAC4 and HDAC5.*

Other HDACi with limited specificity like vorinostat, thicostatin A and belinostat are used and find similar effects on CXCR4 expression.

- 3. AMD3100, a CXCR4 inhibitor.*

Response: We thank this reviewer for carefully reviewing the manuscript and the supportive statement with the constructive comments.

There are only a few concerns that would require attention:

- 1. Major: a) The paper would benefit from information on self-renewing HSC homing/engraftment by providing information on secondary recipient engraftment of homed HSC in primary recipients.*

Response: This comment has now been addressed in the revised manuscript. As requested by the reviewer, we isolated BM of homed primary NSG mice and injected into secondary recipient NSG mice. We found that LMK235-treated CD34⁺ cells also showed increased engraftment in these transplanted secondary recipients (Fig. 2h), suggesting LMK235 treatment promoted homing of self-renewing CB HSCs.

- b) characterization of CXCR4 expression in further refined CD34+ cell subpopulations accounting for self-renewing HSC (CD34+/CD38-/CD90+/CD45RA-/Rh123low/CD49f+) or at the very least CD34+/CD38-/CD90+ cells. As presented, the information is focused on non-HSC but mostly on MPP or ST-HSC.*

Response: We have characterized CXCR4 expression in HSCs (defined by CD34⁺CD38⁻CD45RA⁻CD49f⁺CD90⁺), MPPs (CD34⁺CD38⁻CD45RA⁻CD49f⁻CD90⁻), CD34⁺CD38⁻CD90⁺ and CD34⁺CD38⁻ cells. The results can be found in Fig. 1e, Supplementary Fig. 1c-e and Supplementary Fig. 3c-f.

- 2. Minor: Figure 3: a) Figure 3e would benefit from a histogram plot including the effect of M344.*

Response: We thank the reviewer for this valuable suggestion. As requested by the reviewer, we have added a histogram of M344's effect on Ac-p65 (Fig. 4a).

b) Information on Ac-p65 levels after combinations of M344+C646 and LMK235+C646 to Figure 3 would also be warranted.

Response: We thank the reviewer for this insightful suggestion. We found that treatment of CB CD34⁺ cells with C646 suppressed effects of M344 and LMK235 on p65 acetylation, and we have added the information in the revised manuscript (Fig. 4b).

REVIEWERS' COMMENTS:

Reviewer #1 (Remarks to the Author):

The authors satisfactorily addressed my concerns in the revised manuscript.

Reviewer #2 (Remarks to the Author):

The authors have satisfactorily addressed all of my concerns.

Reviewer #3 (Remarks to the Author):

The authors have responded satisfactorily to my concerns by providing appropriate, well-controlled experiments or additional descriptions of previously described experiments.

We appreciate the time and efforts by the editor and reviewers in reviewing our manuscript and thanks for the valuable and constructive comments.

Following the comments are our point-by-point responses:

** Please shorten all subheadings in the Results and Methods sections to fewer than 60 characters including spaces. Please also ensure that these titles do not contain punctuation.*

Response: This has now been addressed in the revised manuscript. We have shortened all the subheadings accordingly to fewer than 60 characters including spaces.

** In the methods section, please ensure that the dilutions at which each antibody was used is stated, and catalogue numbers are provided for commercial antibodies.*

Response: This has now been included in the revised manuscript.

** Please ensure the references are in the standard Nature format. Please note that dois are required only for online-only publications.*

Response: This has now been addressed in the revised manuscript and four missing dois have been added.

** Please make a statement of competing financial interests after the author contributions section.*

Response: This has now been added in the revised manuscript.

** Figures legends should not exceed 350 words. Please shorten by removing detailed methodological information and/or interpretation, or, if appropriate, consider splitting the affected figures in two. Note that we allow up to 10 display items (figures and tables) in the main manuscript.*

Response: This has now been addressed in the revised manuscript. We have split original Figure 1 and Figure 2 into Figures 1-4. Now the revised manuscript contains 9 figures.

**Please ensure that every Figure panel is described. Please note that the legend for Figure 5 is missing e and f (were these mistakenly labeled I and j?)*

Response: Thanks a lot for pointing out this. This has now been addressed in the revised manuscript.

** Please supply figures as separate vector files with high resolution bitmap*

components.

Response: This has now been addressed in the revised manuscript.

** In each figure where error bars are used, they must be defined, and the number of experimental replicates stated. One statement at the end of each figure is sufficient if the error bars are equivalent throughout the figure.*

Response: This has now been addressed in the revised manuscript.

** Where statistical tests are presented as asterisks, please ensure that the asterisks are defined in each relevant figure legend, together with the name of the statistical test.*

Response: This has now been addressed in the revised manuscript.

** Please ensure that all blots and gels are accompanied by the locations of molecular weight/size markers. Blots should be cropped such that at least one marker position is present. Please also supply uncropped scans of the most important blots as a supplementary figure in the supplementary information. This should be cited once in the Methods section.*

Response: This has now been addressed in the revised manuscript.

** We allow up to 10 display items (figures or tables) in the main text. Please consider whether supplementary items could be transferred to the main manuscript to aid the reader.*

Response: Thank you for your suggestions. The revised manuscript now has 9 figures in the main text.

** Please supply the Supplementary Information as a single, separate PDF file.*

Response: This has now been addressed in the revised manuscript.

** Each supplementary figure should be presented above its corresponding figure legend.*

Response: This has now been addressed in the revised manuscript.

** Please ensure that all novel RNA-seq are deposited in a publicly accessible database, and that accession codes are provided in the Data Availability Statement.*

Response: This has now been addressed in the revised manuscript. The statement

“RNA sequencing data have been deposited in the GEO database under accession number GSE110736.” is provided in the Data Availability.

** Your paper will be accompanied by a two-sentence editor's summary, of between 250-300 characters, when it is published on our homepage. Could you please approve the draft summary below or provide us with a suitably edited version.*

“Enhancement of haematopoietic stem cell (HSC) homing and engraftment is critical for haematopoietic cell transplantation. Here, the authors find that HDAC5 inhibition enhances HSC homing and engraftment by increasing p65 acetylation and enhancing NF-kB mediated CXCR4 transcription.”

Response: We thank the editor for providing this two-sentence summary, which appropriately summarizes our novel findings in this study.

REVIEWERS' COMMENTS:

Reviewer #1 (Remarks to the Author):

The authors satisfactorily addressed my concerns in the revised manuscript.

Response: We thank this reviewer for the positive feedback and the approval of our revised manuscript.

Reviewer #2 (Remarks to the Author):

The authors have satisfactorily addressed all of my concerns.

Response: We are glad to see that this reviewer acknowledged that we have addressed his concerns.

Reviewer #3 (Remarks to the Author):

The authors have responded satisfactorily to my concerns by providing appropriate, well-controlled experiments or additional descriptions of previously described experiments.

Response: We are pleased that this reviewer was satisfied with our revised manuscript.